# Immunopathology and *Trypanosoma congolense* parasite sequestration cause acute cerebral trypanosomiasis

**Sara Silva Pereira[†], Mariana De Niz[†‡], Karine Serre, Marie Ouarné, Joana E Coelho, Cláudio A Franco, Luisa M Figueiredo***

Instituto de Medicina Molecular - João Lobo Antunes, Faculdade de Medicina, Universidade de Lisboa, Lisbon, Portugal

**Abstract** *Trypanosoma congolense* causes a syndrome of variable severity in animals in Africa. Cerebral trypanosomiasis is a severe form, but the mechanism underlying this severity remains unknown. We developed a mouse model of acute cerebral trypanosomiasis and characterized the cellular, behavioral, and physiological consequences of this infection. We show large parasite sequestration in the brain vasculature for long periods of time (up to 8 hr) and extensive neuropathology that associate with ICAM1-mediated recruitment and accumulation of T cells in the brain parenchyma. Antibody-mediated ICAM1 blocking and lymphocyte absence reduce parasite sequestration in the brain and prevent the onset of cerebral trypanosomiasis. Here, we establish a mouse model of acute cerebral trypanosomiasis and we propose a mechanism whereby parasite sequestration, host ICAM1, and CD4[+] T cells play a pivotal role.

**\*For correspondence:**
lmf@medicina.ulisboa.pt

[†]These authors contributed equally to this work

**Present address:** [‡]Trypanosome Cell Biology Unit, Institut Pasteur, Paris, France

## Editor's evaluation

*Trypanosoma congolense* is an important animal trypanosome that exhibits significant biological differences to the better studied *Trypanosoma brucei*. In this study, the authors describe a novel mouse model of cerebral trypanosomiasis, based on the 1/148 strain of *T. congolense*. Using elegant intravital imaging, the authors show that parasites sequester in the vessels of various organs, especially in the brain, causing deleterious T cell responses and inflammation.

## Introduction

Animal welfare is increasingly recognized as a key determinant of Human wellbeing and socio-economic development. This concept of 'One Health' gains additional relevance when tackling infectious diseases endemic to developing countries. Animal African trypanosomiasis, or nagana, a devastating neglected disease endemic to Africa, South America, and Asia, is a great example of this. Nagana costs US$ 1.0–1.2 billion per year in African livestock production losses, but the annual impact on the agricultural Gross Domestic Product is estimated at US$ 4.75 billion (***Food and Agriculture Organization, 2019***). *Trypanosoma congolense* is one of the most prevalent and pathogenic African trypanosome species in Africa (***Bengaly et al., 2002***; ***Gashururu S et al., 2021***; ***Habeeb et al., 2021***; ***Katabazi et al., 2021***). Whilst *T. congolense* infections in African cattle mostly cause a chronic, wasting disease, in exotic breeds and in other mammals, including dogs, goats, and horses, the parasite can cause a rapidly fatal, acute disease, characterized by inflammatory syndrome, disseminated intravascular coagulation syndrome, and neurological impairment (also called cerebral trypanosomiasis) (***Calvet et al., 2020***; ***Griffin and Allonby, 1979***; ***Harrus et al., 1995***; ***Savage et al., 2021***). Currently, the field lacks an animal model that allows the study of acute *T. congolense*-induced trypanosomiasis,

and of cerebral disease as its most severe form. Understanding the determinants of acute trypanosomiasis may help reduce the burden of disease in the long-term.

One of the key aspects that distinguishes *T. congolense* from human-infective *T. brucei* is its ability to cytoadhere to the vascular endothelium rather than to egress the bloodstream and invade tissues (reviewed in *Silva Pereira et al., 2019*). *T. congolense* cytoadhesion causes parasite sequestration (*Losos et al., 1973*; *Losos and Gwamaka, 1973*; *Ojok et al., 2002*), which, for other pathogens, such as *Babesia spp.* and *Plasmodium spp.*, is a key determinant of virulence (*Gallego-Lopez et al., 2019*; *Ghazanfari et al., 2018*; *Rogerson et al., 2007*; *Van den Steen et al., 2013*; *Vargas et al., 2014*). Currently, very little is known about the impact of *T. congolense* sequestration in disease. Yet, parasite presence in the vasculature, and sequestration in particular, usually results in an inflammatory response (*Storm and Craig, 2014*). We know that *T. congolense* adhesion to host cell membranes triggers antibody-complement cascades and increases vascular permeability, suggestive of endothelium damage (*Banks, 1980*). The parasite itself has also been reported to release soluble molecules, like trans-sialidases, that activate the endothelium in vitro, and enhance inflammation in vivo (*Ammar et al., 2013*). In turn, excessive inflammation is a common driver of pathology in many infectious diseases. It is therefore plausible that the physical damage caused by parasite sequestration in the brain and the resulting host's immune response affect disease progression.

Here, we report the first mouse model of acute cerebral trypanosomiasis in animals and investigate its mechanism. We characterized parasite sequestration in the mouse vasculature, the consequences of parasite-endothelial cell interaction, and the drivers of cerebral trypanosomiasis. Our data showed that *T. congolense*-induced cerebral trypanosomiasis is caused by a combination of increased parasite sequestration in the brain and CD4$^+$ T cell activation, via upregulation of intercellular adhesion molecule 1 (ICAM1) expression in endothelial cells. These findings highlight the importance of parasite sequestration and provide the first insights into the cellular mechanism for the development of *T. congolense* cerebral trypanosomiasis in animals.

## Results

### A virulent strain of *T. congolense* causes cerebral trypanosomiasis

We found that infection of C57BL/6 J mice with two independent *T. congolense* strains (1/148 and IL3000) resulted in dramatically different mouse survival. Mouse infection with strain 1/148 resulted in acute disease with mean mouse survival of 9.0±0.4 days (N=4) (*Figure 1A*). The majority of mice did not survive beyond the first peak of parasitemia (*Figure 1B*). After 6 days of infection (1–3 days to the time of death), mice showed growing signs of neurological impairment, including loss of proprioception, hemiparesis (i.e. weakness in one side of the body), strength and grip loss in the limbs, and head enlargement. In contrast, infections with strain IL3000 resulted in 3–5 defined peaks of parasitemia (*Figure 1B*). These mice died within 77.5±4.0 days (*Figure 1A*) with multi-organ and multi-systemic pathology.

We compared organ pathology at the first peak of parasitemia and observed that infection with strain 1/148 resulted in marked damage to the brain and thymus; moderate damage in the liver; and mild damage to the heart, kidneys, lungs, and spleen (*Figure 1C*). Infection with strain IL3000 resulted also in moderate damage to the liver, but only mild damage to the spleen, thymus and testes, and minimal to the heart (*Figure 1C*). Importantly, whilst no pathological changes were apparent in the brain of mice infected with IL3000, 1/148 infections resulted in large brain lesions characterized by multifocal grey matter vacuolation, ischemia, neuronal loss and hemorrhages (*Figure 1D*), which were the most likely cause of death. Parasite labeling by immunohistochemistry showed large parasite accumulation in the brain vasculature (*Figure 1E*), to the point of vascular occlusion in smaller capillaries. Parasite accumulation in the brain vasculature during natural infections has been described in the literature, including in cattle, dogs, horses, and wild animals, some of which have been reported to develop neurological impairment (*Losos et al., 1973*; *Losos and Gwamaka, 1973*; *Savage et al., 2021*).

Two out of seven mice infected with strain 1/148 showed marked or moderate damage to the kidneys. To investigate whether kidney failure could contribute to early death in 1/148-infected mice, we assessed creatinine, urea, and Neutrophil gelatinase-associated lipocalin (NGAL) in the serum of mice infected with either strain (*Figure 1F*). We observed increased levels of urea in both strains,

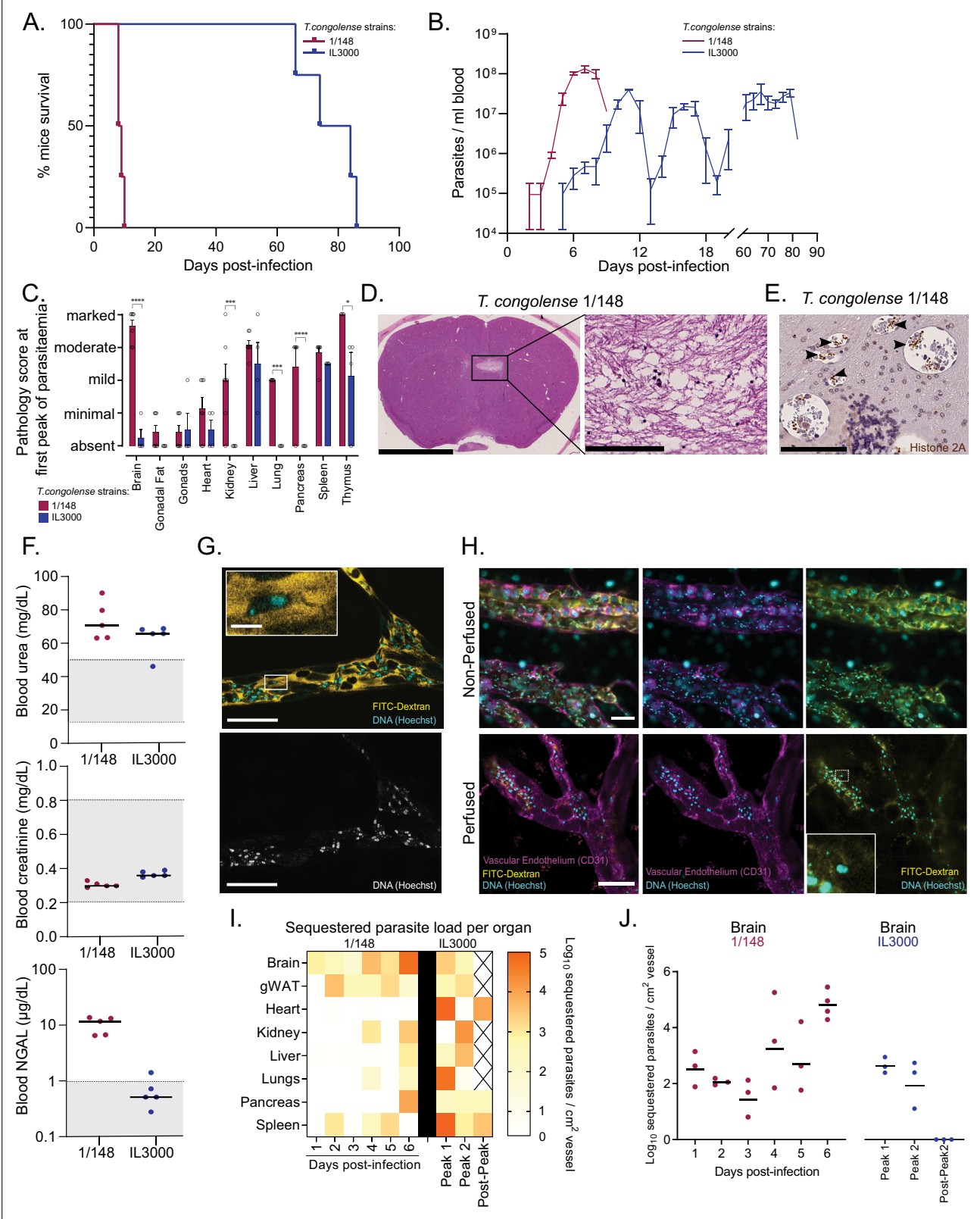

**Figure 1.** Infection progression and phenotypic differences between *T. congolense* savannah strains IL3000 (*Gibson, 2012*) and 1/148 (*Young and Godfrey, 1983*). (**a**). Mice survival curves following infection with *T. congolense* strains 1/148 and IL3000 (average of 9.0±0.4 and 77.5±4.0 (Mean ± SEM) days respectively) (N=4). (**b**). Parasitemia throughout infection with 1/148 and IL3000 parasites estimated by hemocytometry shows that infections with strain 1/148 do not progress past the first peak of parasitemia, whereas parasitemia with strain IL3000 oscillates from below the level of detection to 5

*Figure 1 continued on next page*

*Figure 1 continued*

× 10⁷ parasites/mL for up to 84 days. Mean ± SEM (**c**). Pathological report at the first peak of parasitemia of each strain, scored based on the degree of organ damage from absent to marked. Stars indicate statistically significant results; two-way ANOVA with Sidak's multiple comparisons test, * p<0.05; ** p<0.01; *** p<0.001; **** p<0.0001. N=4–7, 2 independent experiments. (**d**). Right: Representative histological hematoxylin and eosin staining of brain of a mouse infected with strain 1/148, showing a large lesion (Scale bar = 2.5 mm). Right: high-magnification of lesion, showing cell loss (scale bar = 100 µm) (**e**). Immuno-histochemical staining of trypanosome H2A (brown), showing parasite sequestration in the brain vasculature. Scale bar = 100 µm (**f**). Serum levels of urea, creatinine, and neutrophil gelatinase-associated lipocalin (NGAL) in mice infected with either T. congolense strains 1/148 or IL3000, at the first peak of parasitemia. Grey area represents normal range for non-infected, healthy mice. (**g**). Representative image of *T. congolense* parasites in a vessel of the brain by intravital microscopy at day 6 post-infection. 70 kDa FITC-Dextran is used for vascular flow labeling (shown in yellow); cell nuclei are stained with Hoechst (shown in cyan). Bottom panel shows nuclei staining only. Scale = 20 µm. On top-left, a zoomed-in section showing a *T. congolense* parasite. Scale = 4 µm. (**h**). Representative images of T. congolense 1/148 parasites in brain vessels by intravital microscopy at day 6 post-infection, without (top) and with (bottom) perfusion. Seventy kDa FITC-Dextran is used for vascular flow labeling (shown in yellow); cell nuclei are stained with Hoechst (shown in cyan), vascular endothelium is stained with α-A637-CD31. Scale = 30 µm. On bottom-right, a zoomed-in section showing a sequestered *T. congolense* parasite after perfusion. (**i**). Heatmap showing sequestered parasite load per organ over the course of the infection in strains 1/148 and IL3000, quantified by intravital microscopy and represented as log₁₀ sequestered parasites per cm² of vessel, adjusted for the vascular density of each organ. (**J**). Sequestered parasite load in the brain vasculature over the course of the infection in strains 1/148 and IL3000, quantified by intravital microscopy.

The online version of this article includes the following figure supplement(s) for figure 1:

**Figure supplement 1.** Parasite load and sequestration in the vasculature of mice infected with *T. congolense* 1/148 and IL3000, at day 6 post-infection.

suggestive of mild dehydration, but normal levels of creatinine, suggestive of normal kidney function. Furthermore, we detected increased levels of NGAL in 1/148-infected mice only. NGAL is a biomarker for the early detection of kidney injury (*Soni et al., 2010*). Together, kidney biochemistry analyses show that, at the first peak of parasitemia, 1/148-infected mice show early signs of kidney damage, whereas IL3000 do not. Regardless, kidney function has not yet been compromised, so kidney failure is not the most likely cause of death. In contrast, the severity of lesions observed in the CNS of 1/148-infected mice and the large accumulation of parasites in the brain microvasculature strongly support that C57BL/6 J mice infected by *T. congolense* 1/148 die of cerebral trypanosomiasis.

## *Trypanosoma congolense* 1/148 preferentially sequesters in the brain vasculature

Given the severity of lesions in the host and the large accumulation of parasites in the brains of animals infected with the *T. congolense* strain 1/148, we hypothesized that cerebral trypanosomiasis was caused by parasite sequestration in the brain vasculature.

Therefore, we started by confirming that *T. congolense* parasites cytoadhered to the endothelium by intravital microscopy. Since we did not have access to fluorescent 1/148 parasites, we used FITC-Dextran to visualize the intravascular environment, Hoechst dye to mark the nuclei of circulating cells (*Figure 1G*), and α-A637-CD31 to stain the vascular endothelium (*Figure 1H*). Hoechst dye stains the nuclei of both host cells and parasites. Within the confines of the endothelium, this includes mostly immune cells. Intravascular parasites and host cells are clearly distinguishable by their size and shape (cellular and nuclear), as well as by the presence of a kinetoplast in the parasites (*Video 1*). So, this combination of markers allowed us to quantify parasites and leukocytes in vessels. A 3D reconstruction is shown in *Video 2*. Then, based on parasite

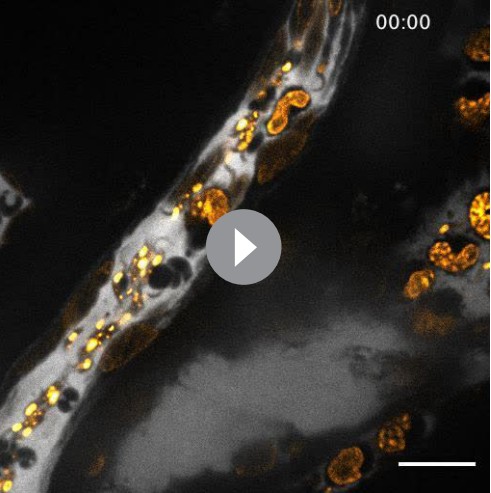

**Video 1.** Intravital imaging of mouse brain microvasculature at day 6 post-infection with *T. congolense* 1/148, showing differentiation between parasites and leukocytes. Imaging was done in a non-perfused mouse, under blocked flow. DNA is stained with Hoechst (yellow); intravascular environment is stained with FITC-Dextran (black and white).
https://elifesciences.org/articles/77440/figures#video1

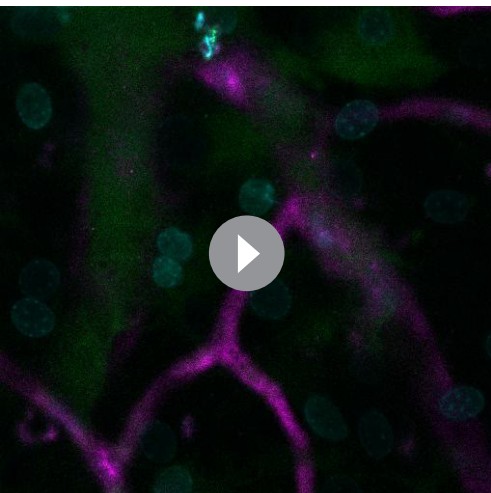

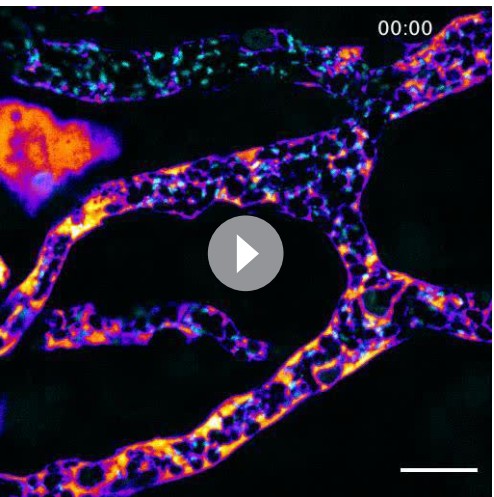

**Video 2.** 3D reconstruction (Z-stack) of brain vasculature at day 6 post-infection with *T. congolense* 1/148. DNA is stained with Hoechst (cyan); intravascular environment is stained with FITC-Dextran (green); vascular walls are labelled with anti-CD31 (purple). https://elifesciences.org/articles/77440/figures#video2

**Video 3.** Intravital imaging of mouse brain microvasculature at day 6 post-infection with *T. congolense* 1/148. Imaging was done in a non-perfused mouse, under unblocked flow. DNA is stained with Hoechst; intravascular environment is stained with FITC-Dextran.
https://elifesciences.org/articles/77440/figures#video3

mobility during live imaging, we compared total parasite load and the percentage of sequestered parasites in major organs in the first 6 days of 1/148 infection and at three points in IL3000 infection (first parasitemia peak, second peak, and post-second peak). For representative image acquisition, we perfused mice to remove parasites not sequestered (*Figure 1H*).

Using these videos, first we compared the total number of parasites (sequestered or freely swimming) in the vessels of multiple organs across the two parasite strains. In 1/148 infections, parasite load increased linearly with time in the brain ($R^2$=0.83, Pearson's correlation) and adipose tissue ($R^2$=0.68, Pearson's correlation) (*Figure 1—figure supplement 1A*, left). In remaining organs, parasite load oscillated during infection. For instance, in the liver, parasite load was maximal already on day 2; in the spleen, maximum parasite load was reached on day 5 p.i; and in the lungs on day 4 p.i. The brain was the organ with the highest parasite load at day 6 p.i., and variability across replicates was low (*Figure 1—figure supplement 1A*, left, *Video 3*). This was also the maximum parasite load registered throughout the course of infection amongst all organs. In IL3000 infections, parasite load was highest in the lungs, at both peaks of parasitemia (1.25–8.64 x $10^5$ parasites / $cm^2$ vessel), followed by heart (1.14–2.15 x $10^5$ parasites / $cm^2$ vessel) and brain (0.61–2.16 x $10^5$ parasites / $cm^2$ vessel). Parasite distribution from the first to second peaks of parasitemia did not change drastically, but there was a 2-fold decrease in the spleen (*Figure 1—figure supplement 1A*, right). After the second peak of parasitemia, in a window of undetectable peripheral parasitemia by hemocytometry, there was a small number of parasites detectable in the brain, heart, lungs, pancreas, and spleen.

Next, we analyzed parasite sequestration. First, we observed that parasite sequestration did not increase linearly with parasite load in neither 1/148 nor IL3000 infections (*Figure 1—figure supplement 1B*). Then, we used the parasite load and the percentage of sequestration to estimate the number of sequestered parasites per organ throughout infection (*Figure 1I*). We observed that parasites of strain 1/148 preferentially sequestered in the brain, reaching ~$10^5$ sequestered parasites/$cm^2$ vessel (*Figure 1I and J*). In contrast, strain IL3000 sequestered mainly in the heart, lungs, and spleen in the first peak of parasitemia and in a less tissue-specific manner in the second peak (*Figure 1I*). Representative images of *T. congolense* sequestration in the different organs at the first peak of parasitemia (day 6 p.i. for 1/148 and days 7–10 p.i. for IL3000 infections) are shown in *Figure 1—figure supplement 1C*. Examples of parasites sequestered imaged under unblocked flow are given in *Videos 4–6*.

Together, these data showed that parasite sequestration is tissue-, strain-, and time-dependent, but not a consequence of high parasitemia. In the *T. brucei* model of cerebral trypanosomiasis,

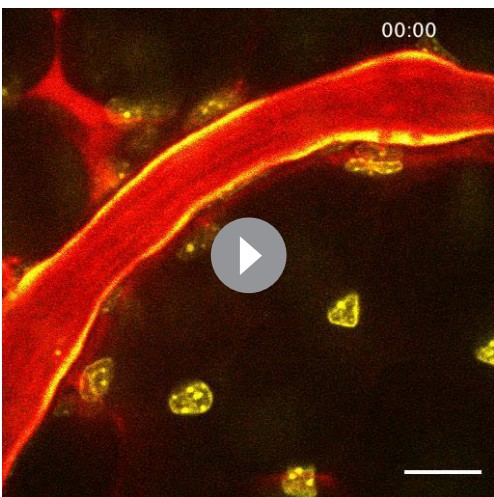

**Video 4.** Intravital imaging of a mouse pancreatic arteriole at day 6 post-infection with *T. congolense* 1/148. Imaging was done in a non-perfused mouse, under unblocked flow. DNA is stained with Hoechst; intravascular environment is stained with FITC-Dextran.
https://elifesciences.org/articles/77440/figures#video4

**Video 6.** Intravital imaging of mouse pancreatic microvasculature at day 6 post-infection with *T. congolense* 1/148, showing parasite motility under slow flow. Imaging was done in a non-perfused mouse, under unblocked flow. DNA is stained with Hoechst (cyan); intravascular environment is stained with FITC-Dextran (yellow).
https://elifesciences.org/articles/77440/figures#video6

there is also no correlation between the levels of blood parasitemia and levels of brain invasion (*Laperchia et al., 2016*). Importantly, given that the brain was the organ with the highest total and sequestered parasite load at day 6 after 1/148 infection, it supported the hypothesis that parasite sequestration in the brain contributes to cerebral trypanosomiasis and early death.

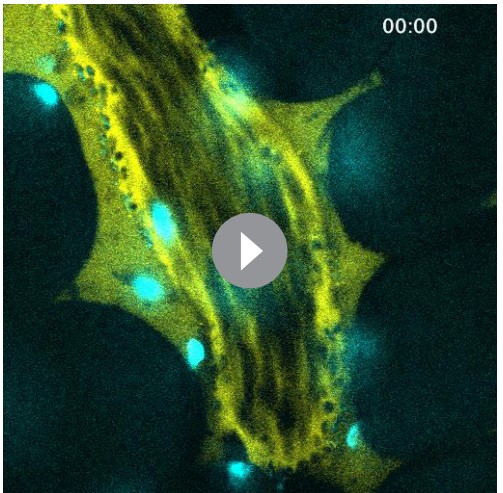

**Video 5.** Intravital imaging of mouse brain microvasculature at day 6 post-infection with *T. congolense* 1/148, showing parasite motility under fast flow. Imaging was done in a non-perfused mouse, under unblocked flow along the vascular endothelium. Parasites can be observed moving against flow. DNA is stained with Hoechst (cyan); intravascular environment is stained with FITC-Dextran (yellow).
https://elifesciences.org/articles/77440/figures#video5

## Individual parasites damage the vascular endothelium and remain sequestered in the brain vasculature for up to 8 hr

We sought to investigate how *T. congolense* parasites of strain 1/148 establish tropism to the brain. Therefore, we used ex-vivo microscopy to understand whether they accumulated in particular brain regions or brain vessels. Mice were infected and, at the first peak of parasitemia, brains were surgically removed, dissected into 10 anatomical regions, and imaged immediately. We found that, in both 1/148 and IL3000 infections, parasites accumulated in the posterior parts of the brain (i.e. cerebellum, midbrain, pons, medulla) (*Figure 2A, B*, left). However, while IL3000 sequestration reproduced this pattern, 1/148 sequestration was evenly distributed across vessels of all brain sections (*Figure 2A, B*, right). Next, we questioned whether vessel caliber was a determinant of parasite sequestration, so we compared the number of parasites in capillaries ($\varphi < 10$ μm), and arterioles/venules of different diameters ($10 \leq \varphi > 20$ μm, $20 \leq \varphi > 40$ μm, $\varphi \geq 40$ μm). We found that *T. congolense* parasites

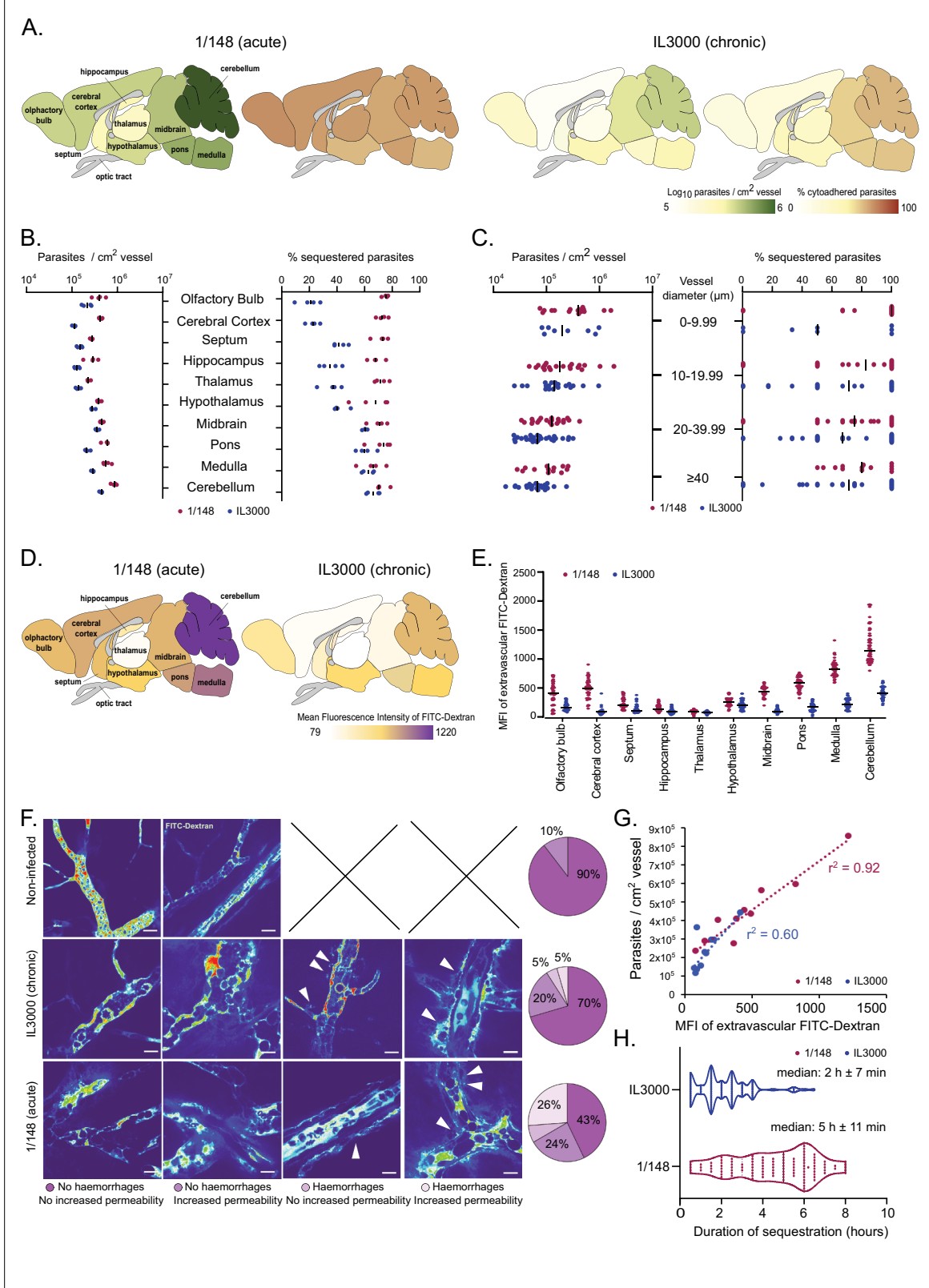

**Figure 2.** Distribution and characterization of Trypanosoma congolense sequestration in the brain. (**a**). Total parasite load (green) and percentage of parasite sequestration (orange) in different parts of the brain at the first peak of parasitemia of strains 1/148 and IL3000, quantified by ex-vivo microscopy in a minimum of 100 independent images in three to four independent mice. (**b**). Quantitative values corresponding to (**a**) at the first peak of parasitemia. (**c**). Parasite load (left) and sequestration proportion (right) in vessels of different diameters. (**d**). Mean fluorescence intensity in

*Figure 2 continued on next page*

Figure 2 continued

the extravascular area of different parts of the brain at the first peak of parasitemia of strains 1/148 and IL3000, quantified by intravital and ex-vivo microscopy in a minimum of 100 independent images in three to four mice, independently infected. (**e**). Quantitative values corresponding to (**d**) at the first peak of parasitemia. Values were measured within 10 μm of the nearest vessel. (**f**). Percentage of brain area displaying four types of vascular pathology (or lack thereof): no hemorrhages and no increased permeability; no hemorrhages and increased permeability; hemorrhages and no increased permeability; hemorrhages and increased permeability and their representative images. (**g**). Correlation between parasite sequestration and vascular permeability as measured by FITC-Dextran MFI. (**h**). Time (in hr) that individual parasites remain cytoadhered. Images were obtained every 30 min, for 12 hr. Black line indicates the mean. N=3, 3 independent infections, 100 vessels per mouse.

distributed evenly (*Figure 2C*, left) and sequestered in similar proportions throughout vessels of all diameters (*Figure 2C*, right).

During the imaging sessions, we detected that the 70 kDa FITC-Dextran diffused differently from the brain vasculature depending on the *T. congolense* strain used for infection. By quantifying FITC-Dextran extravascularly, we were able to assess changes in vascular permeability upon infection. We observed that infection with strain 1/148 results in increased vascular permeability, particularly in the posterior regions of the brain (*Figure 2D, E*). We also detected and quantified the presence of microhemorrhages in the brain vasculature (*Figure 2F*). Furthermore, particularly in 1/148 infections, vascular permeability in the brain increased linearly with parasite load ($r^2$=0.92, Pearson's correlation) (*Figure 2G*). We observed that the vessels with increased permeability and/or evidence of microhemorrhages were more common (57% vs 30%) in the acute (1/148) than chronic (IL3000) infections, which further supports the hypothesis that 1/148 infection cause cerebral trypanosomiasis.

Next, we followed 100 parasites for up to 10 hr in vivo and discovered that despite regular blood flow, 1/148 parasites remained adhered to the same endothelial cell to a maximum of 8 hr, and for longer than IL3000 (median of 2 hr±7 min and 5 hr±11 min for IL3000 and 1/148, respectively) (*Figure 3H*). Whilst we observed that parasites may attach to the brain endothelium with other parts of their cell, we confirmed that they often adhere with their distal flagellum (*Video 7*), as previously reported for parasites colonizing the mesentery (*Banks, 1978*).

In summary, we observed that parasites of both strains preferentially colonized the vasculature of the posterior parts of the brain, but that 1/148 sequestered evenly across brain regions, corroborating our earlier observation that sequestration was independent of parasite load. These experiments also showed very little physical constraints for parasite distribution and sequestration, as parasite distribution and sequestration occur in vessels of all calibers. Importantly, we showed that 1/148 sequestration in the brain causes more damage to the vasculature than IL3000 and that the parasite-endothelial cell interaction is longer-lived, thus corroborating the tropism of this strain to the brain vasculature.

The analysis described so far comparing IL3000 and 1/148 strains clearly indicates that *T. congolense* 1/148 infection in C57BL/6 J mice is a good model of acute cerebral trypanosomiasis. For the remainder of this study, we investigated the causes of the development of cerebral trypanosomiasis in mice.

## *T. congolense* 1/148 infection induces a pro-inflammatory profile in brain endothelial cells

Given the prolonged and frequent cell-to-cell contact between parasites and endothelial cells, we started by characterizing how the brain endothelium responded to 1/148 infection. To achieve this, we infected RiboTag.PDGFb-iCRE mice, which, upon induction of Cre recombinase activity, express a hemagglutinin tag at the ribosomes of the endothelial cells. We harvested the brain, captured the polysomes of the endothelial cells by immunoprecipitation, and sequenced the mRNA that was in translation (*Sanz et al., 2009*; *Figure 3A*). We compared the expression profiles of brain endothelial cells of mice non-infected and infected with *T. congolense* 1/148 and found that infection resulted in downregulation of 588 genes and upregulation of 612 genes (*Figure 3B* and *Supplementary file 1*).

Within the 20 most downregulated genes, we found a variety of functions, including genes involved in metabolism, efflux transport, and two genes involved in maturation of B cells (lymphocyte antigen 6 complex, locus C1 (Ly6c1) and tumor necrosis factor receptor superfamily, member 17 (Tnfrsf17m also called BCMA); *Figure 3C*). Within the 20 most upregulated transcripts, we detected several genes involved in the immune response, and in particular those related to type 1 immunity (i.e. CXCL10, CXCL9, Basic leucine zipper transcription factor (Batf2), Growth differentiation factor

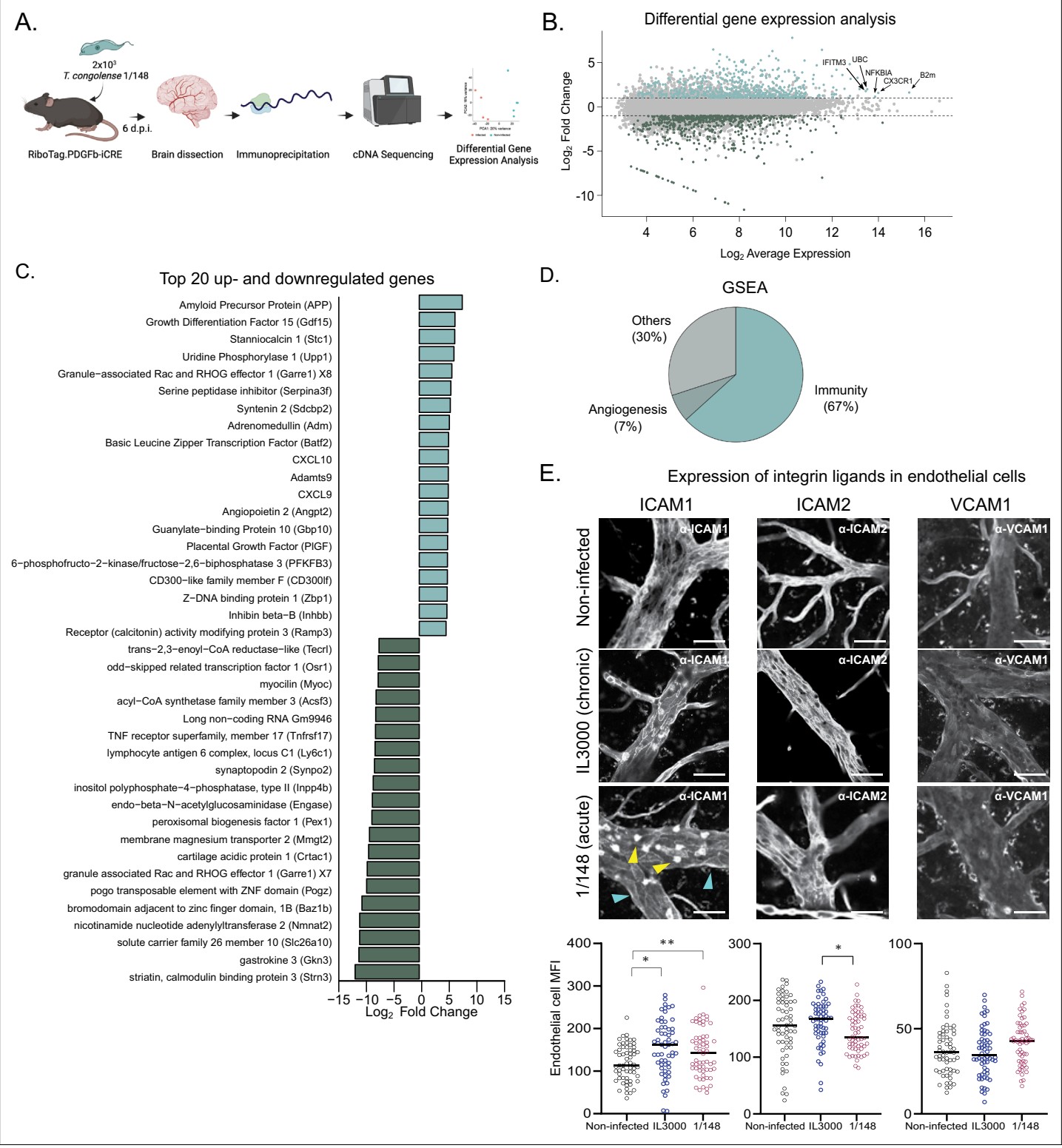

**Figure 3.** T. congolense infection induces a pro-inflammatory profile in brain endothelial cells. (**a**). Schematics of the methodology used to compare the transcriptomes of brain endothelial cells. RiboTag.PDGFb.iCRE mice, under Cre recombinase induction, were infected with $2 \times 10^3$ T. congolense 1/148 parasites intra-peritoneally. At day 6 post-infection, mice were euthanized, perfused and brains were dissected and homogenized. The polysomes of endothelial cells were immunoprecipitated and the RNA extracted and converted into cDNA, which was sequenced on a NextSeq 500 platform as 75 bp single-end reads (N=4–5). Figure created with https://biorender.com/. (**b**). Distribution of detected transcripts in terms of average expression and fold change between infected and non-infected conditions. Transcripts upregulated in infection are shown in red (N=612), downregulated transcripts

*Figure 3 continued on next page*

*Figure 3 continued*

are shown in blue (N=588). The top most abundant upregulated transcripts are identified: beta-2 microglobulin (B2m), chemokine (C-X3-C motif) receptor 1 (Cx3cr1), nuclear factor of kappa light polypeptide gene enhancer in B cells inhibitor, alpha (Nfkbia), ubiquitin C (UBC), and interferon induced transmembrane protein 3 (Ifitm3). (**c**). The 20 most up- and downregulated transcripts detected upon infection, ordered by log$_2$ fold change. (**d**). Enriched gene sets upon infection (FDR < 0.05) detected by gene ontology-based gene set enrichment analysis (***Subramanian et al., 2005***), produced by WEBgestalt (***Wang et al., 2017***), and ordered by the normalized enriched score. (**e**). Mean fluorescent intensity of ICAM1, ICAM2, and VCAM in the brain endothelium in non-infected mice and in mice infected with T. congolense 1/148 or IL3000, at the first peak of parasitemia, and respective representative images. Black lines represent mean. Stars indicate statistically significant results; repeated measures one-way ANOVA with Tukey's multiple comparisons test, * p<0.05; ** p<0.01; *** p<0.001; **** p<0.0001.

The online version of this article includes the following figure supplement(s) for figure 3:

**Figure supplement 1.** Expression of endothelial cell surface integrin ligands ICAM1, ICAM2, and VCAM1 at the first peak of infection with either *T. congolense* 1/148 or IL3000.

15 (Gdf15), CD300-like family member F (CD300lf), Z-DNA binding protein 1 (Zbp1), and guanylate-binding protein 10 (Gbp10)). We also detected transcripts involved in angiogenesis, endothelial transmigration, and in the response to blood-brain-barrier dysfunction, including amyloid precursor protein (APP), stanniocalcin 1 (Stc1), serine peptidase inhibitor (Serpina3f), syntenin 2 (Sdcbp2), angiopoietin 2 (Angpt2), and placental growth factor (Pgf) (***Figure 3C***).

To study the predicted biological function of upregulated genes in a more systematic and unbiased way, we performed gene set enrichment analysis of the 612 genes (GSEA) (***Supplementary file 2***). Results were consistent with the signature we observed from the most upregulated genes, as 63% of the enriched gene sets related to inflammatory responses (e.g. response to interferon gamma, positive regulation of defense response, response to tumor necrosis factor, cytokine-mediated signaling pathway, regulation of innate immune response, response to virus, response to interleukin-1, antigen processing and presentation, regulation of inflammatory response, leukocyte migration, etc.), and 7% to angiogenesis (e.g. angiogenesis, regeneration) (***Figure 3D***).

These results show that *T. congolense* 1/148 infection results in endothelial activation, which is consistent with previous results (***Ammar et al., 2013***). So, and given that an activated endothelium upregulates the expression of integrin ligands which control the immune response (***Lawson and Wolf, 2009***), we also assessed the expression changes of intercellular adhesion molecule (ICAM) 1, ICAM2,

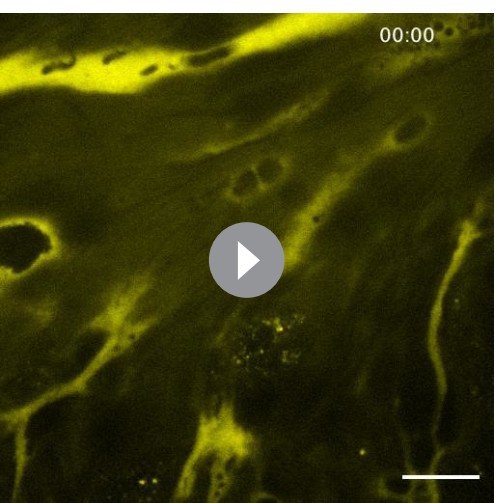

**Video 7.** Intravital imaging of mouse brain microvasculature at day 6 post-infection with *T. congolense* 1/148, showing parasite attached to the endothelial cell with the distal flagellum. Imaging was done in a non-perfused mouse, under unblocked flow. DNA is stained with Hoechst; intravascular environment is stained with FITC-Dextran.
https://elifesciences.org/articles/77440/figures#video7

and vascular cell adhesion molecule (VCAM) 1, three integrin ligands involved in leukocyte transendothelial migration. We compared expression of ICAM1, ICAM2, and VCAM1 proteins in the vasculature of the brain at the first peak of parasitemia in both strains, by ex-vivo microscopy (***Figure 3F***). ICAM1 was highly expressed in immune cells (yellow arrowheads, ***Figure 3E***), but also detectable in vessel walls (cyan arrowheads, ***Figure 3E***). Consistent with the endothelium ribosome profiling data (which revealed ICAM1 to be upregulated upon 1/148 infection (FC = 1.62), but not of ICAM2 or VCAM1), we observed increased expression of ICAM1 in the endothelium of the brain (unpaired t-test with Welch's correction, p-value = 0.0312), but not of ICAM2, nor VCAM1 (***Figure 3G***). In IL3000 infections, we also observed an increase in ICAM1 expression by endothelial cells (p-value = 0.003). To determine whether the increase in endothelial ICAM1 was brain-specific, we checked ICAM1, ICAM2, and VCAM1 expression in remaining organs. During 1/148 infections, we observed ICAM1 also increased in the lungs, heart, and liver (unpaired t-test with Welch's correction, p-value < 0.01),

whereas ICAM2 expression increased in the heart and kidneys (unpaired t-test with Welch's correction, p-value < 0.0001), and VCAM1 in the liver, spleen, and kidneys (unpaired t-test with Welch's correction, p-value < 0.01) (*Figure 3—figure supplement 1* A, left). During IL3000 infections, we did not observe increased ICAM1 expression in remaining organs (Figure S2A, right). Furthermore, we observed that leukocytes also expressed increased ICAM1 (fivefold) and ICAM2 (twofold), but not VCAM1, upon infection with either strain.

Similar to what we did to assess the distribution of parasite load and sequestration, we asked whether the increase in ICAM1 expression was localized to a particular region in the brain. We observed that, during 1/148 infections, expression of ICAM1 increased in all anatomic regions of the brain, with the exception of the hypothalamus (*Figure 3—figure supplement 1B, C*). In contrast, during IL3000 infections, ICAM1 expression increased in all brain regions with the exception of the hippocampus, hypothalamus, and medulla. Given that parasite sequestration remains even throughout brain despite variations in endothelial ICAM1 expression suggests that ICAM1 is not the sole receptor for parasite cytoadhesion.

In summary, analysis of the brain vasculature revealed that endothelial cells respond to *T. congolense* infection by displaying a pro-inflammatory response. To further investigate the development of *T. congolense* 1/148-associated neuropathology, we compared the dynamics of immune cell subsets in the brain of mice infected with either strain.

## *T. congolense* 1/148 infection, but not IL3000, results in increased numbers of intravascular ICAM1+ monocytes and extravascular CD4+ T cells in the brain

Next, we characterized the immune response in the systemic circulation at the first peak of parasitemia of both infections by serum cytokine profiling and flow cytometry. We assessed the serum levels of six cytokines (i.e. IL-1α, IL-1β, IFNγ, CXCL10, CXCL9, TNFα) at the first peak of parasitemia of both 1/148 and IL3000 infections. The transcripts of these cytokines were shown to be upregulated in endothelial cells (*Figure 3C*). We confirmed a significant increase in the levels of CXCL9 and CXCL10 at days 3 and 6 p.i., during both 1/148 and IL3000 infections, compared to the non-infected control (twoway ANOVA, CXCL9: p-value < 0.0091 and=0.0107, respectively; CXCL10: p-value = 0.0013 and 0.0082; *Figure 4A*). The levels of TNFα show a tendency to increase with infection in both strains. However, the levels of IFNγ only show an ascending trend during 1/148 infections. These results support the presence of a systemic pro-inflammatory response in the first week of infection with both strains of *T. congolense*.

In terms of leukocyte cell subsets, we observed no significant changes in the number of myeloid cells (i.e. dendritic cells, monocytes and macrophages) in either strain (*Figure 4B*). Given the immunofluorescence data supporting an increase in ICAM1 expression by leukocytes upon infection, we compared the numbers of ICAM1 expressing leukocytes in the blood. We observed that the number of ICAM1+ monocytes in the blood during 1/148 infections increased 25-fold compared to the non-infected control and 4-fold compared to IL3000 infections (two-way ANOVA with Tukey's Multiple Comparison Test, p-value = 0.0002 and 0.0131) (*Figure 4C*).

Then, we characterized the leukocyte populations within the brain. To distinguish and quantify the number of intra- and extravascular leukocytes, we administered α-CD45-APC antibody intravenously, 3 min prior to mouse euthanasia. In such a short incubation time, the antibody does not traverse the vasculature and thus only stains intravascular leukocytes (*Anderson et al., 2014*; *Morawski et al., 2017*). *Figure 4D* confirms that the α-CD45-APC antibody was efficient at staining intravascular leukocytes. We found an increase in the number of both extravascular (unpaired t-test, p-value = 0.0482, t=2.813, df = 4) and intravascular (unpaired t-test, p-value = 0.0160, t=4.0007, df = 4) CD45+ cells upon 1/148 infection, but not upon IL3000 infection (*Figure 4—figure supplement 1*), suggesting that the immune response is much stronger in the acute strain. Additionally, the contribution of intravascular leukocytes to the total immune cells in the brain is higher in infected animals (14%±4 *vs* 25%±1).

Leukocytes are a heterogenous population of cells. To characterize the changes within the myeloid (e.g. monocytes, macrophages and dendritic cells) and lymphoid (B and T cells) subpopulations, we stained brain leukocytes for various subset markers, and quantified them using flow cytometry (*Figure 4—figure supplement 2*). We did not stain for neutrophils because preliminary experiments

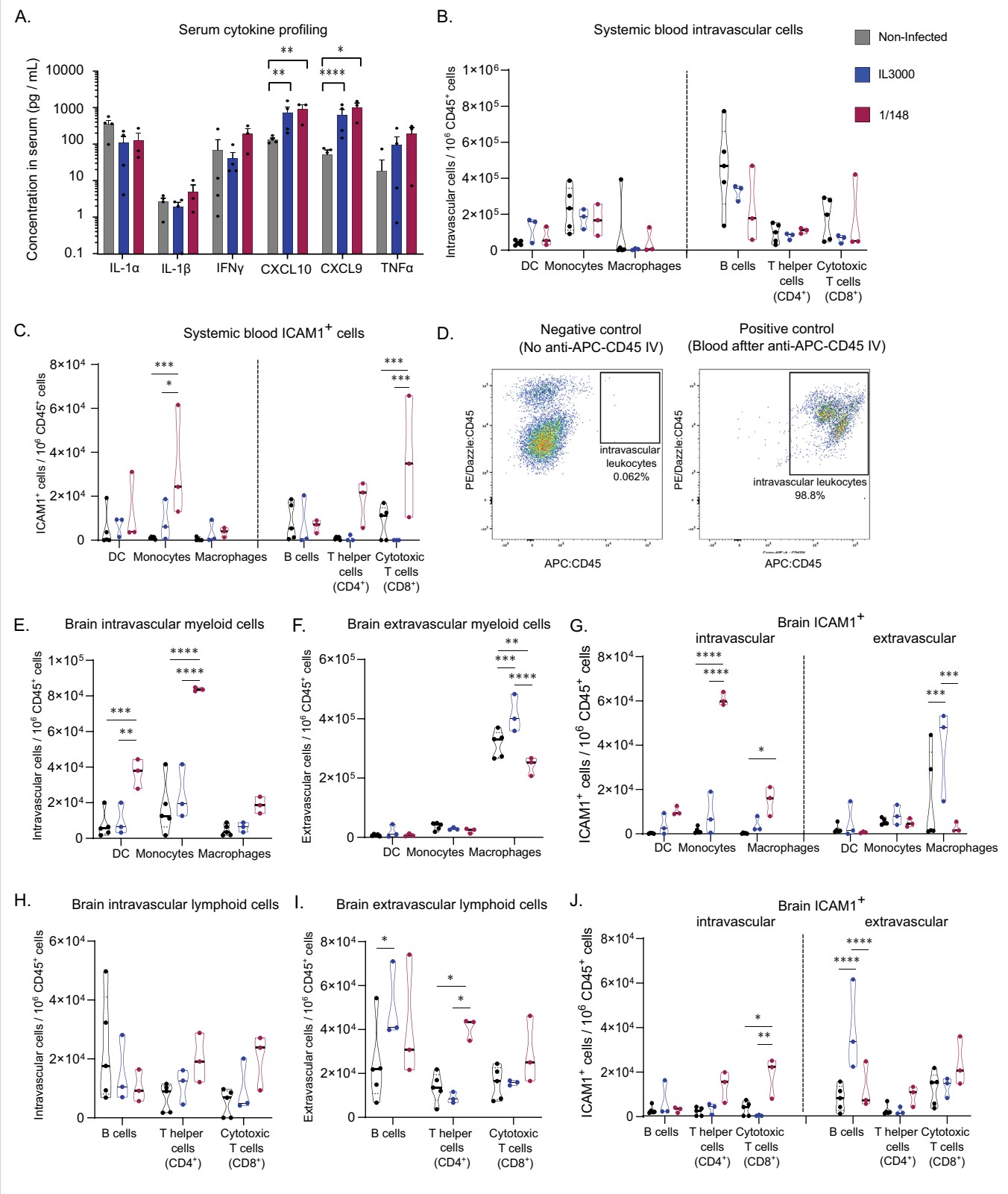

**Figure 4.** Characterization of immune response in the systemic blood and brain at the first peak of parasitemia post-infection with T. congolense 1/148 or IL3000. (**a**). Serum concentration of IL-1α, IL-1β, IFNγ, CXCL10, CXCL9, TNFα in non-infected mice and mice infected with either 1/148 or IL3000, at the first peak of parasitemia (Mean ± SEM; N=3–4). (**b**). Number of myeloid (CD11b+) and lymphoid (CD3+, CD19+) cells per mL of blood, separated by cell type (CD11c+ Dendritic Cells, Ly6C+ Monocytes, and F4/80 Macrophages, CD19+ B cells, CD3+CD4+ T cells, CD3+CD8+ T cells) in non-infected and

*Figure 4 continued on next page*

*Figure 4 continued*

infected mice. (**c**). Number of leukocytes (CD45$^+$) expressing ICAM1 per mL of blood, separated by cell type, in non-infected and infected mice. (**d**). Left: Representative graph of brain cells from an infected mouse that did not receive APC-CD45 intravenous injection, showing no cells on the APC-CD45 gate (negative control). Right: Representative image of blood cells from an infected mouse that received APC-CD45 intravenous injection, showing that, as expected, almost all cells were APC-CD45+ (positive control). (**e**). Number of intravascular (APC:CD45$^-$) myeloid (CD11b$^+$) cells per brain of mice, separated by cell type in non-infected and infected mice. (**f**). Number of extravascular (APC:CD45$^+$) myeloid (CD11b$^+$) cells per brain of mice, separated by cell type, in non-infected and infected mice. (**g**). Number of intravascular (APC:CD45$^-$) and extravascular (APC:CD45$^+$) myeloid (CD11b$^+$) cells expressing ICAM1 per brain of mice, separated by cell type, in non-infected and infected mice. (**h**). Number of intravascular (APC:CD45$^-$) non-myeloid (CD11b$^-$) cells per brain of mice, separated by cell type in non-infected and infected mice. (**i**). Number of extravascular (APC:CD45$^+$) non-myeloid (CD11b$^-$) cells per brain of mice, separated by cell type, in non-infected and infected mice. (**j**). Number of intravascular (APC:CD45$^-$) and extravascular (APC:CD45$^+$) non-myeloid (CD11b$^-$) cells expressing ICAM1 per brain of mice, separated by cell type, in non-infected and infected mice. Stars indicate statistically significant results; unpaired t-test, * p<0.05; ** p<0.01; *** p<0.001; **** p<0.0001. Black line indicates mean. N=3.

The online version of this article includes the following figure supplement(s) for figure 4:

**Figure supplement 1.** Analysis of extra and intravascular leukocytes in the brain.

**Figure supplement 2.** Gating strategy for the characterization of systemic and brain leukocyte populations.

revealed that neutrophil numbers do not change significantly upon infection (Figure S3B). The number of intravascular CD11b$^+$Ly6C$^-$CD11c$^+$ dendritic cells (DC) and CD11b$^+$Ly6C$^+$ monocytes increased by fourfold in 1/148 infections only (two-way ANOVA, p-value = 0.0005, <0.0001, respectively) (*Figure 4E*). As expected, the numbers of extravascular DCs and monocytes was low, but the number of macrophages was high in both infected and non-infected conditions (*Figure 4F*). Importantly, 73% of the intravascular monocytes expressed ICAM1 during 1/148 infection, compared to only 8% in the non-infected control and 32% in IL3000 infection, which is reflected in a significant increase (35-fold) in the number of ICAM1-expressing monocytes circulating in the brain vasculature (two-way ANOVA, p-value < 0.0001) (*Figure 4G*).

The number of intravascular lymphoid cells showed only a tendency to increase, in both strains (*Figure 4H*). In contrast, the number of extravascular lymphoid cells present in the brain changed differently depending on the strain. IL3000-infected mice had increased numbers of extravascular B cells in the brain parenchyma (two-way ANOVA, p-value = 0.0341), whereas in the 1/148 infection, T helper cells (CD4$^+$) were increased by twofold (two-way ANOVA, p-value = 0.0.0305; *Figure 4I*). In terms of ICAM1 expression, intravascular cytotoxic T cells (CD8$^+$) were increased in 1/148 infections (two-way ANOVA, p-value = 0.0120) and extravascular ICAM1$^+$ B cells were increased in IL3000 infections (two-way ANOVA, p-value < 0.0001) (*Figure 4J*).

Overall, these data show that, upon infection, the development of cerebral trypanosomiasis is associated with the recruitment of inflammatory monocytes to the brain vasculature, most of which express ICAM1 (*Figure 4E*), and the infiltration of CD4$^+$ T cells into the brain parenchyma (*Figure 4F*).

## Blocking ICAM1 reduces disease severity and parasite sequestration in the brain

ICAM1 appears to be an important regulatory molecule in *T. congolense* infection. On one hand, it is upregulated in the brain endothelial cells upon infection with both strains of *T. congolense*. On the other hand, the upregulation in inflammatory monocytes in the brain vasculature was only detected in infections with the virulent strain (1/148), suggesting that ICAM1 may play a role in disease severity, perhaps by favoring parasite sequestration and/or lymphocyte recruitment. We found previously that *T. brucei* interacts with several endothelial receptors (*De Niz et al., 2021*). To test if *T. congolense* sequestration depends on similar host molecules, we blocked ICAM1 in vivo, by administering α-ICAM1 antibody 24 hr before infection, repeating daily during the course of infection. We noticed that α-ICAM1-treated mice showed fewer signs of cerebral disease, so we used SHIRPA (i.e. Smith-Kline Beecham, Harwell, Imperial College, Royal London Hospital, phenotype assessment), a mouse phenotypic assessment protocol (*Rogers et al., 1997*), to test whether α-ICAM1-treated mice were neurologically fitter than isotype-treated controls.

We selected five parameters of the semi-quantitative SHIRPA protocol: posture, velocity of escape upon touching, positional passivity, type of locomotion, and grip strength. Mice infected with the strain 1/148 showed defects in all these parameters on day 6 post-infection (*Figure 5A*). These alterations are consistent with the changes described in dogs infected with *T. congolense* (*Harrus et al.,*

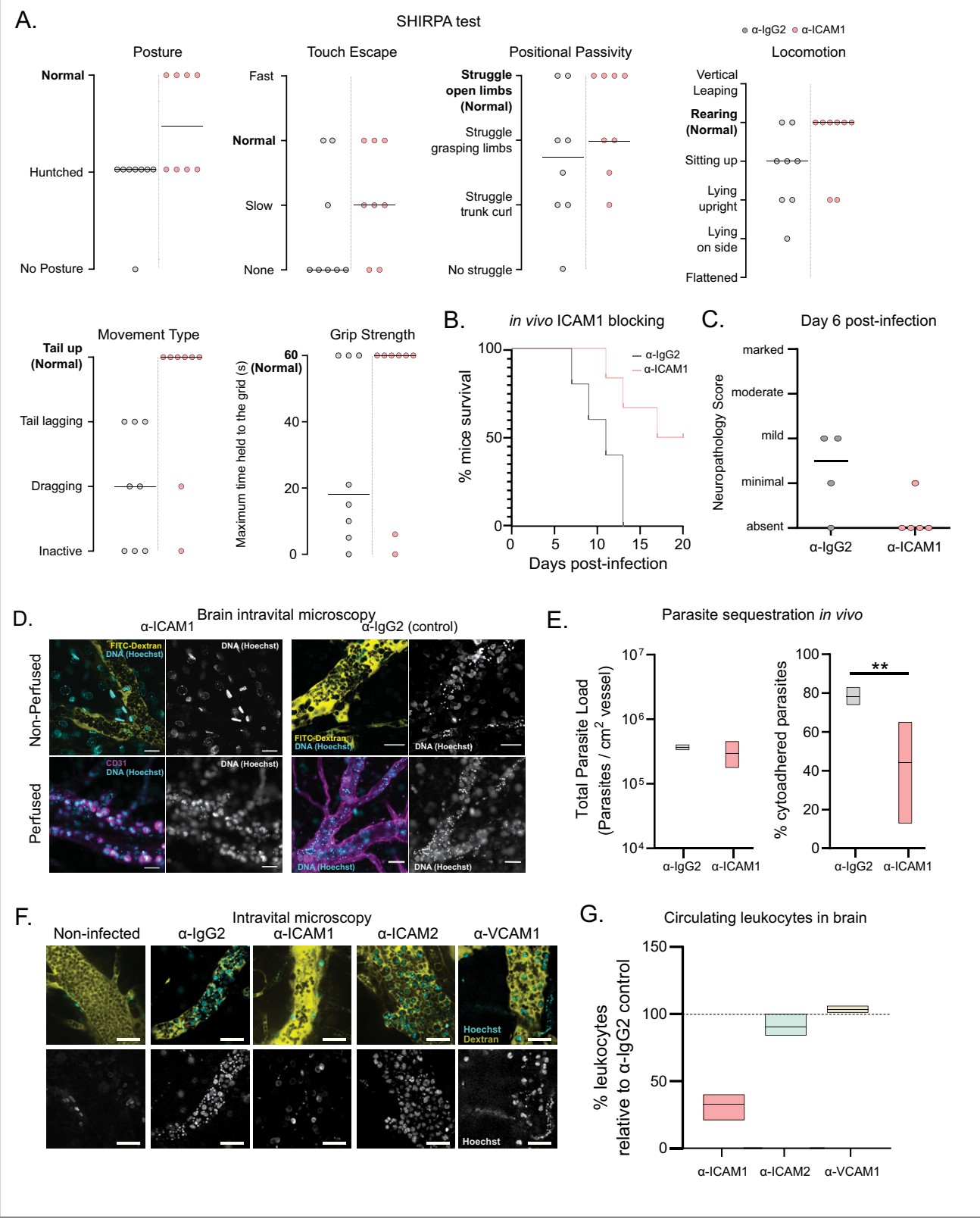

**Figure 5.** Phenotypic effects of anti-ICAM1 antibody treatment in mice infected with T. congolense 1/148. (**a**). Behavioral assessment of mice treated with either α-ICAM1 antibody or its isotype control (α-IgG2), based on the SHIRPA protocol. The parameters assessed and shown are: Posture, Touch Escape, Positional Passivity, Locomotion, Movement Type (and tail posture), and Grip Strength: Time (in s) that mice can hold upside down on a grid (maximum of 1 min). Black line indicates median. (**b**). Mice survival curves following treatment with α-ICAM1 or its isotype control α-IgG2. Results were

*Figure 5 continued on next page*

*Figure 5 continued*

pooled from two independent experiments (N=6). (**c**). Neuropathology score at day 6 post-infection of mice infected with T. congolense 1/148, treated with α-ICAM1 antibody or its isotype control (N=4–5). Black line indicates median. (**d**). Representative images obtained by intravital microscopy, of parasite sequestration in the brain vasculature upon treatment with α-ICAM1 antibody or its isotype control with and without perfusion, at day 6 post-infection. Scale bar = 30 μm. (**e**). Parasite load (left) and percentage of parasite sequestration (right) in the brain of mice at day 6 post-infection upon treatment with α-ICAM1 antibody or its isotype control, quantified by intravital microscopy (N=3 in 3 independent infections). Stars indicate statistically significant results; mixed models, ANOVA, multiple comparisons test, * $p<0.05$; ** $p<0.01$; *** $p<0.001$; **** $p<0.0001$. (**f**). Representative images obtained by intravital microscopy, of leukocytes in the brain vasculature in non-infected mice, and mice infected with T. congolense 1/148, but treated with antibodies against IgG2 (isotype control), ICAM1, ICAM2, or VCAM1. Nuclei are stained with Hoechst dye and shown in green. (**g**). Percentage of nucleated cells (leukocytes) found circulating in the brain vasculature at day 6 post-infection in mice treated with α–ICAM1, ICAM2, or VCAM1, relative to the isotype control (represented as 100%). Black line indicates mean. (N=3 in 3 independent infections).

The online version of this article includes the following figure supplement(s) for figure 5:

**Figure supplement 1.** Quantification of parasite binding to recombinant ICAM1 protein in vitro.

*1995*), showing that SHIRPA protocol is a useful quantitative method to score neurological alterations associated with cerebral trypanosomiasis in this mouse model. The SHIRPA protocol revealed that α-ICAM1-treated mice performed better than the control group in all parameters relating to neurological impairment, and particularly those relating to movement and strength (*Figure 5A*). Importantly, we also found that α-ICAM1-treated mice survived longer than isotype-treated controls, as 50% of the mice were still alive at day 20 p.i, when the experiment was terminated (*Figure 5B*). Brain necropsy also showed reduced neuropathology (*Figure 5C*).

To understand the cellular basis of this ICAM1-dependent phenotype, we quantified parasite load and sequestration percentage in the brain at day 6 p.i., by intravital microscopy (*Figure 5D*). We found that total parasite load was unaffected by treatment (*Figure 5E*, left), but the percentage of sequestered parasites was reduced by 44% (two-way ANOVA, p-value = 0.0025) (*Figure 5E*, right). These results could suggest that ICAM1 directly promotes parasite sequestration. To test if ICAM1 is a potential receptor for parasite adhesion, we immobilized recombinant ICAM1, CD36 and BSA on a plastic surface, in static conditions, and measured 1/148 parasite adherence by microscopy after a 30-min incubation (*Figure 5—figure supplement 1A*). Parasites bound equally efficiently to recombinant ICAM1 protein, to BSA or recombinant CD36 protein (*Figure 5—figure supplement 1*). Whilst we cannot exclude the possibility that by collecting parasites from the blood, we are selecting those with lower sequestration capacity or lower affinity to ICAM1, these results suggest that *T. congolense* 1/148 parasite sequestration in static conditions is not enhanced by ICAM1 binding.

These data show that blocking ICAM1 reduces parasite sequestration in the brain, improves cerebral trypanosomiasis severity and promotes mouse survival, but this effect appears to be independent of direct parasite binding to the ICAM1 receptor. Given that ICAM1 is important in leukocyte recruitment and infiltration (*Lawson and Wolf, 2009*), next we investigated whether the phenotypic effect of ICAM1 blocking could derive from an impaired immunological response. First, we measured the impact of ICAM1 blocking in the number of circulating nucleated cells specifically in the brain vasculature. By intravital imaging, we detected a 67% ± 5% reduction in the number of circulating nucleated cells (assumed to be leukocytes due to their size and complexity; one-way ANOVA, p-value = 0.0270), which was not observed when blocking either ICAM2 or VCAM1 (*Figure 5F and G*).

Overall, these results indicate that the ICAM1 blocking may improve disease outcome by reducing parasite sequestration and the magnitude of the immune response.

## Cerebral trypanosomiasis is prevented in the absence of T cells

Immunophenotyping of 1/148 infected mice showed an increase in both ICAM1+ intravascular monocytes and extravascular CD4+ T cells. To understand if any of these cell types could be the cause of cerebral trypanosomiasis, we performed two experiments to individually deplete each of these cell types. If one of the cell types contributes to pathology, its depletion should lead to reduced disease signs and prolonged host survival. To deplete circulating monocytes and inflammatory macrophages, we administered clodronate liposomes intravenously 24 hr prior and 3 days post-infection. We observed that mice treated with clodronate liposomes showed high peripheral parasitemia earlier than mice receiving either PBS-filled liposomes or PBS only (*Figure 6—figure supplement 1A*). Survival was also affected. Mice receiving clodronate liposomes died between

days 6 and 7, compared to days 7–8 when receiving control liposomes, and days 7–10 if only PBS was administered (*Figure 6—figure supplement 1B*). Moreover, we performed mouse behavioral assessment on day 6 post-infection and observed that monocyte/macrophage depletion increased severity of neurological impairment (*Figure 6—figure supplement 1C*). These results show that neuropathology is not caused by monocytes/inflammatory macrophages. In fact, these cells have a protective effect during infection. Furthermore, this suggests that the role of ICAM1 in cerebral disease is not dependent on innate immune cells alone, but rather on the increased expression by endothelial cells.

Then, we tested the role of lymphoid cells in neuropathology. In other parasitic diseases, like cerebral malaria, brain immunopathology is caused by small numbers of infiltrating CD8+ T cells (*Belnoue et al., 2002*). Moreover, upregulation of ICAM1, IFNγ, and related cytokines is known to increase T cell adhesion and transendothelial cell migration (*Sonar et al., 2017*; *May and Ager, 1992*; *Sancho et al., 1999*). So, to investigate the role of lymphoid cells in parasite sequestration and pathology in the brain, we used mice of the same genetic background, but that do not produce mature T and B cells (RAG2 KO mice). Although they also lack mature B cells, mice with cerebral trypanosomiasis do not survive past the first peak of parasitemia, so the antibody-mediated response is unlikely to play an important role in ICAM1-mediated neuropathology. Besides, we did not observe a significant variation in the number of B cells in the brain upon 1/148 infection (*Figure 4F–G*).

So, we assessed several infection parameters in RAG2 KO mice and compared them to WT mice. First, we performed intravital microscopy in the brain of RAG2 KO mice infected with *T. congolense* 1/148 at day 6 p.i. (*Figure 6A*) and quantified the number of circulating nucleated cells. We observed that it was reduced relative to WT mice and similar to the levels observed in α-ICAM1-treated WT mice (*Figure 6B* and *Figure 5G*). Then, we followed disease progression, and observed that RAG2 KO mice performed better than WT in every parameter of the SHIRPA test, with only one mouse (out of five) showing signs of neurological impairment (*Figure 6C*). As expected due to the absence of B cells, mice did not clear parasitemia, and remained heavily parasitized, with minor oscillations (*Figure 6D*). Remarkably, parasitemia was not increased compared to WT controls, and mice survived up to day 40 p.i. (*Figure 6E*). Brain necropsy corroborated these observations as the level of brain lesions was lower overall, despite one mouse still showing moderate neuropathology (*Figure 6F*). These data suggest that neuropathology associated with cerebral trypanosomiasis may be caused by T cells.

Given our previous results indicating a role of ICAM1 in cerebral trypanosomiasis, we investigated whether T cells could cause neuropathology via ICAM1 signaling. Therefore, we administered α-ICAM1 or its isotype control into RAG2 KO mice and infected them with *T. congolense* 1/148 parasites. At day 6 p.i., α-IgG2-treated RAG2 KO mice had similar parasite load in the vasculature of the brain to α-IgG2-treated WT (*Figure 7G*), but presented lower levels of parasite sequestration (two-way ANOVA, p-value = 0.0241) (*Figure 6H*). This phenotype is similar to what we observed in α-ICAM1-treated WT mice (*Figure 5E*). Furthermore, we observed that α-ICAM1-treatment caused no effect on RAG2 KO mice in terms of both parasite load and percentage of parasite sequestration (*Figure 7G and H*). The fact that the effect of α-ICAM1 treatment on WT mice is mimicked in untreated RAG2 KO mice shows that, in the absence of B and T cells, there is no additive role for ICAM1 in parasite sequestration or cerebral trypanosomiasis.

In summary, we conclude that T cells are the cause of cerebral trypanosomiasis and that ICAM1 acts upstream them.

## CD4+ T cells are sufficient to induce cerebral trypanosomiasis

To understand which T cell subset was responsible for neuropathology and exclude the unlikely role of B cells in the development of cerebral trypanosomiasis, we performed an adoptive transfer experiment, where we restored the B cell, CD4+ or CD8+ T cell populations of naïve RAG2 KO (*Figure 7A*). Live lymphoid cells were collected from wild type mice, separated by fluorescence-activated cell sorting (*Figure 7A and B*), and injected intravenously into RAG2 KO mice. One-week post injection, we confirmed engrafting of the cell populations by flow cytometry. Following infection with 1/148 parasites, we observed that the mice receiving CD4+ T cells, but not the remaining experimental groups, succumbed to disease, indicating that CD4+ T cells are sufficient to induce cerebral trypanosomiasis. These results show that in the acute cerebral trypanosomiasis infection model, mice succumb rapidly to infection of 1/148 strain due to toxic effect of T helper cells.

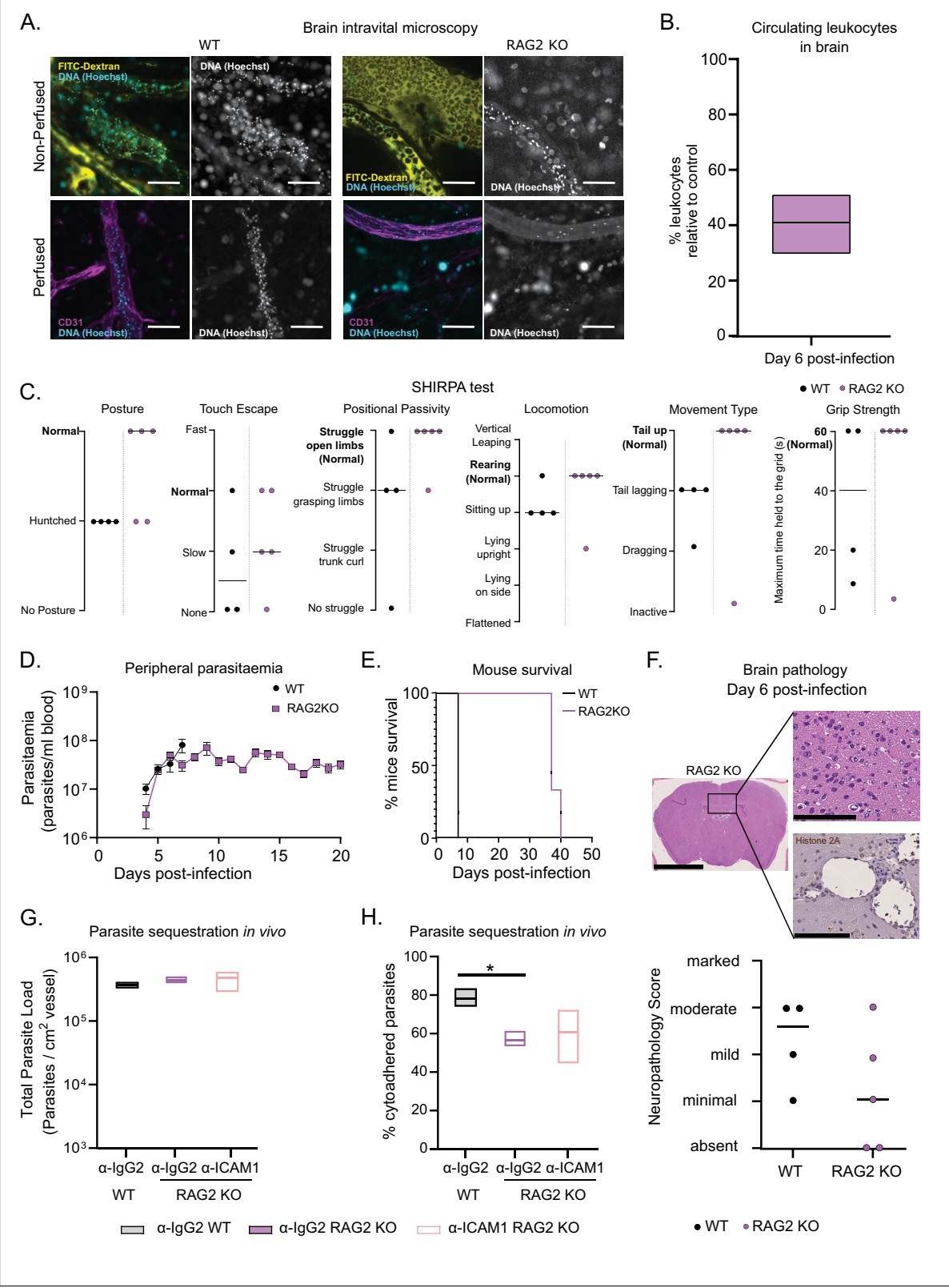

**Figure 6.** *T. congolense* 1/148 sequestration and infection progression in RAG2 KO mice. (**a**). Representative images of parasite sequestration in brains of WT and RAG2 KO mice with and without perfusion, at day 6 post-infection. Scale bar = 30 μm. (**b**). Percentage of leukocytes found in the brain vasculature at day 6 post-infection in RAG2 KO mice relative to WT control. (**c**). Behavioral assessment of mice treated with either a-ICAM1 antibody or its isotype control (a-IgG2), based on the SHIRPA protocol. Black line indicates median. (**d**). Parasitemia curve of T. congolense 1/148 in WT or RAG2 KO

*Figure 6 continued on next page*

*Figure 6 continued*

mice, measured by hemocytometry. Mean ± SEM (**e**). Mice survival curves of WT and RAG2 KO mice (N=4). (**f**). Top: Neuropathology score at day 6 post-infection of RAG2 KO and WT mice infected with *T. congolense* 1/148 (N=4–5). Black line indicates median. Bottom right: Representative histological hematoxylin nd eosin staining of brain of a RAG2 KO mouse infected with strain 1/148, showing absence of lesions in the brain parenchyma (Scale bar = 2.5 mm). Top right: high-magnification of brain parenchyma (scale bar = 100 μm). Bottom right: Immuno-histochemical staining of trypanosome H2A (brown) showing little parasite sequestration in the brain vasculature. Scale bar = 100 μm. (**g**). Parasite load in the brain of WT or RAG2 KO mice, treated with either α-IgG2 or α-ICAM1 antibody, quantified by intravital microscopy, at day 6 post-infection. (**h**). Percentage of cytoadhered parasites in the brain of WT or RAG2 KO mice, treated with either α-IgG2 or α-ICAM1 antibody, quantified by intravital microscopy. Stars indicate statistically significant results; two-way ANOVA, multiple comparisons test, * $p<0.05$; ** $p<0.01$; *** $p<0.001$; **** $p<0.0001$. (N=3 in 3 independent infections).

The online version of this article includes the following figure supplement(s) for figure 6:

**Figure supplement 1.** Effect of monocyte/macrophage depletion on *T. congolense* 1/148 infections.

## Discussion

Animal African trypanosomiasis comprises a spectrum of diseases caused by multiple organisms. In natural infections, *T. congolense* is the most prevalent species, yet its mechanisms of disease remain poorly explored. In this work, we established the first mouse model of *T. congolense* cerebral trypanosomiasis and uncovered a role of trypanosome sequestration in disease severity. We show that C57BL/6 J mice infected with *T. congolense* 1/148 strain show significant and quantifiable neurological clinical signs, which are associated with prolonged parasite sequestration in the brain vasculature and an overwhelming ICAM1-mediated, pro-inflammatory CD4$^+$ T cell response.

In the acute disease caused by strain 1/148, the brain is the site of preferential parasite sequestration. The presence of *T. congolense* has previously been shown to result in endothelium activation via the release of soluble factors in vitro (*Ammar et al., 2013*). Given our observations of *T. congolense* 1/148 parasites accumulating in the brain vasculature and having a prolonged interaction with the endothelium, we propose that parasite sequestration enhances endothelial activation in vivo. It is worth noting that IL3000 parasites still populate the brain vasculature in similar levels as 1/148 and they also trigger inflammatory cascades. However, most of those parasites are non-sequestered forms and we showed that the inflammatory response is very distinct from that of cerebral trypanosomiasis. Tropism to the brain vasculature is a virulence factor and is a common feature of intravascular parasites. Trypanosome sequestration on the brain endothelium seems to cause the vascular occlusion that results in ischemia and tissue hypoxia, accounting for some of the pathological lesions we observed in infected mice. In both cerebral malaria and acute babesiosis, attenuation of strain virulence is accompanied by loss of cerebral capillary sequestration (*Dondorp et al., 2004*; *Medana and Turner, 2006*; *Sondgeroth et al., 2013*), much like what we observed when comparing *T. congolense* strains 1/148 and IL3000. In extravascular parasites, such as *T. brucei*, brain tropism is also a virulence factor, albeit displayed by extravasation rather than sequestration (*Grab and Kennedy, 2008*).

Ultimately, the fact that RAG2 KO mice, but not monocyte/macrophage-depleted mice, survive four times longer than WT, and that CD4$^+$ T cell reconstitution is sufficient to restore acute disease, shows that T helper cells drive *T. congolense* cerebral trypanosomiasis. Interestingly, T cell activation, regardless of CD4 or CD8 expression, is essential for the development of *T. brucei* cerebral trypanosomiasis in humans because it allows parasite crossing of the blood-brain-barrier by temporarily opening the tight junctions of the endothelial cells (*Laperchia et al., 2016*; *Olivera et al., 2021*). However, it is worth noting that this disease, also called second-stage sleeping sickness, only develops later in infection, and not as a manifestation of acute disease like what we report here for *T. congolense*.

T cell suppression by cyclosporine A administration has been shown not to alter disease course nor associated inflammation in chronic models of trypanosomiasis (*Noyes et al., 2009*). In contrast, we would expect that the same treatment in a 1/148 infection would result in prolonged survival, mimicking the effect we observed in RAG2 KO mice. Moreover, although ICAM1 is upregulated in endothelial cells both in vivo (*Figure 3G*) and in vitro (*Ammar et al., 2013*) in chronic IL3000 infections, it does not result in life-threatening neuroinflammation (*Figure 1C*). This suggests that, whilst the exacerbated immune response may be the cause of neuropathology, 1/148 parasite tropism to the brain vasculature by sequestration seems to be the differential factor for the development of cerebral trypanosomiasis.

Based on our observations, we propose a model of acute cerebral trypanosomiasis that can serve as a basis for future studies (*Figure 8*). We propose that parasite sequestration to endothelial cells results

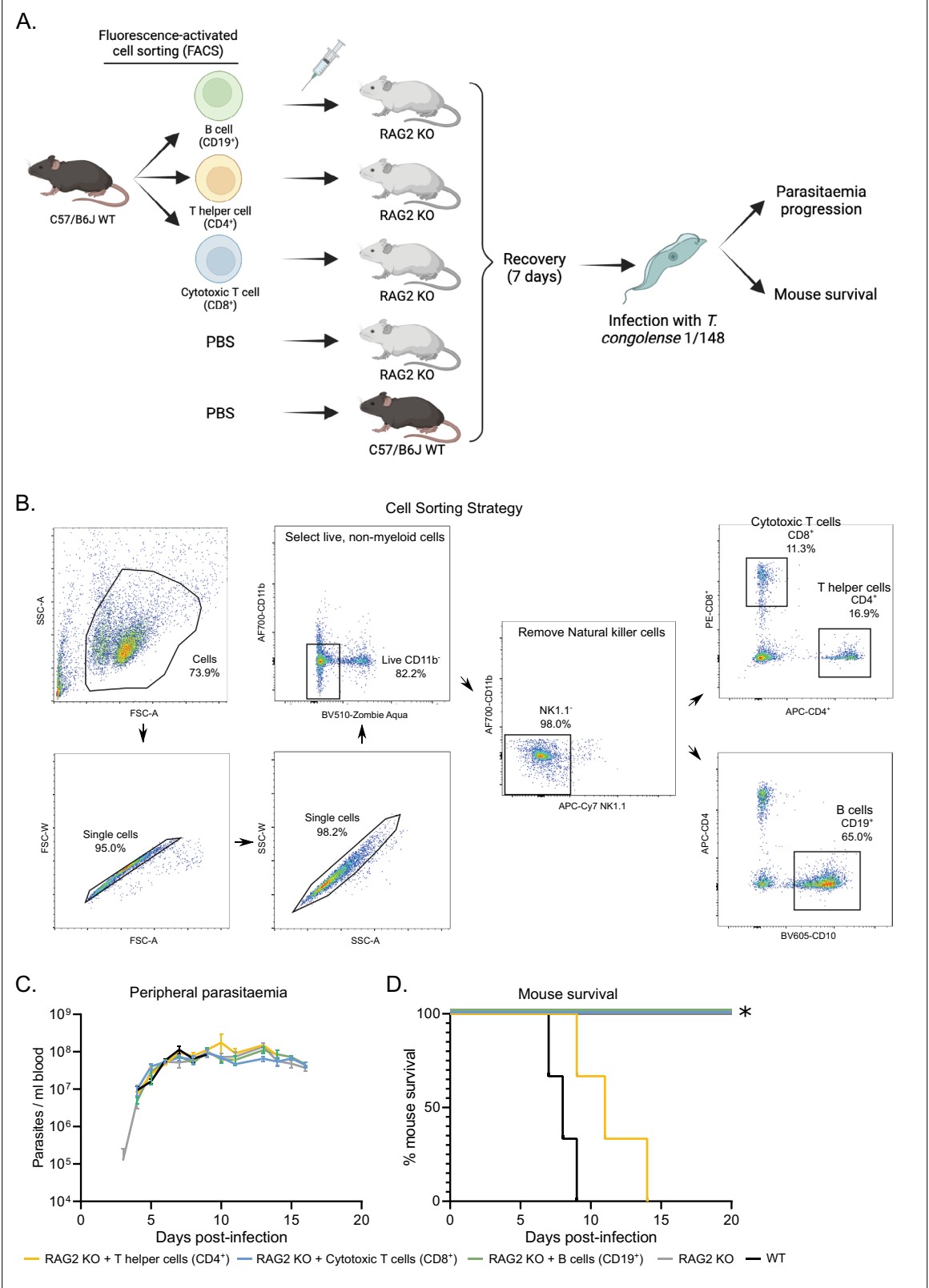

**Figure 7.** Phenotypic effects of adoptive lymphoid cell transfer into RAG2 KO mice before infection with T. congolense 1/148. (**a**). Schematics depicting experimental procedure. Cells were harvested from the spleens and lymph nodes of three C57BL/6 J wild-type mice and stained for different lymphoid cell subsets. (B) cells (CD19+), T helper cells (CD4+), and cytotoxic T cells (CD8+) were sorted and collected by flow cytometry, and injected intravenously into naïve RAG2 KO mice. Two control groups comprising either RAG2 KO mice or wild-type mice received a similar volume of vehicle. Mice were

*Figure 7 continued on next page*

*Figure 7 continued*

allowed to recover for 7 days, after which they were infected with *T. congolense* 1/148. Infection progression was assessed daily. Figure created with https://biorender.com/. (**b**). Gating strategy used for fluorescence-activated cell sorting, in order to select pure B cells, T helper cells and cytotoxic T cells. (**c**). Parasitemia curve of mice infected with *T. congolense* 1/148, estimated by hemocytometry. (**d**). Mouse survival following infection with T. congolense 1/148. Experiment was terminated at day 20 post-infection. N=3.

in activation of endothelial cells. Previously, this activation has been shown to occur as a response to soluble molecules, such as the parasitic trans-sialidades, via the NF-κB pathway (*Ammar et al., 2013*). Endothelial cell activation is displayed by an increase in ICAM1 expression and the release of pro-inflammatory cytokines. These could lead to the increase in circulating myeloid cells in the brain, particularly of ICAM1[+] monocytes. Some of the monocytes cross the blood-brain-barrier, accumulate in the brain and differentiate into macrophages. These cytokines, chemokines and myeloid cells can promote CD4[+] T cell activation and recruitment to the brain vasculature (*Clark et al., 2007*). The interaction of T cells with ICAM1 at the surface of endothelial cells, and possibly myeloid cells, is likely to facilitate activated CD4[+] T cell adhesion and extravasation to the brain parenchyma (*Dietrich, 2002*), ultimately causing the neuropathology that culminates in early death.

The similarities of the *T. congolense*-associated neuropathology with that of cerebral malaria are clearly remarkable, but perhaps not unexpected if we reconsider that the initial events that trigger the immune response, namely sequestration and endothelial damage, are alike. ICAM1 is one the endothelial cell adhesion receptors for *P. falciparum* (*Berendt et al., 1989*), but has a multifaceted role in cerebral malaria pathogenesis (*Storm and Craig, 2014*). Whilst we did not find evidence for ICAM1 to be an endothelial cell receptor for *T. congolense* parasites, this hypothesis cannot be excluded until more sensitive and complex adhesion assays are performed. Both CXCL9 and CXCL10 play important roles in the inflammation that drives cerebral malaria-associated neuropathology, namely in terms of T cell adhesion to endothelial cells (*Sorensen et al., 2018*), and recruitment to the brain parenchyma (*Campanella et al., 2008*). Monocytes promote brain inflammation, secretion of IFNγ, and activation of CD8[+] T effector cells (*Schumak et al., 2015*).

The prevalence of cerebral trypanosomiasis is small in current disease settings. However, whilst cerebral trypanosomiasis is not currently common in cattle, it is described in dogs, goats, and horses (*Calvet et al., 2020*; *Griffin and Allonby, 1979*; *Harrus et al., 1995*; *Savage et al., 2021*). Until now, its disease mechanism could not be experimentally addressed because we lacked a representative animal model. Furthermore, mitigating acute disease may allow the introduction in Africa of exotic cattle breeds that are more profitable, thus contributing to the socioeconomic development of African countries and the reduction of poverty. The lack of suitable prophylaxis and/or treatment for trypanosomiasis has resulted in a severe reduction in livestock farming in tsetse-endemic areas. Despite being less fit for farming (i.e. reduced size, meat and milk yield), the very few cattle heads being raised are African breeds (*Bos indicus*), which have increased tolerance to trypanosomiasis (*Berthier et al., 2015*; *d'Ieteren et al., 1998*). Introduction of external, perhaps more economically-appealing breeds is avoided because they could rapidly succumb to trypanosomiasis. This was very clearly observed upon the introduction of *T. vivax* in South America. Although *T. vivax* mostly causes a chronic infection in Africa (with the single exception of the *T. vivax* hemorrhagic fever), in Brazil it results in frequent epidemics of acute disease, with mortality rates that can reach 70% (*Batista et al., 2012*; *Cadioli et al., 2012*; *Garcia et al., 2016*). The difficulty in rearing non-native breeds in Africa is an enormous bottleneck for economic development.

To conclude, we present a new experimental model that adds new tools to understanding trypanosomiasis and its spectrum of clinical outcomes.

## Materials and methods
### Animal xperiments
This study was conducted in accordance with EU regulations and ethical approval was obtained from the Animal Ethics Committee of Instituto de Medicina Molecular (AWB_2016_07_LF_Tropism). Infections were performed at the rodent facility of Instituto de Medicina Molecular, in 6–10 weeks old, wild-type, male C57BL/6 J mice (Charles River, France), RiboTag.PDGFb-iCRE mice, or RAG2 KO mice of the same age, bred in-house. Mice were infected by intraperitoneal injection (i. p) of $2 \times 10^3$ [*T.*

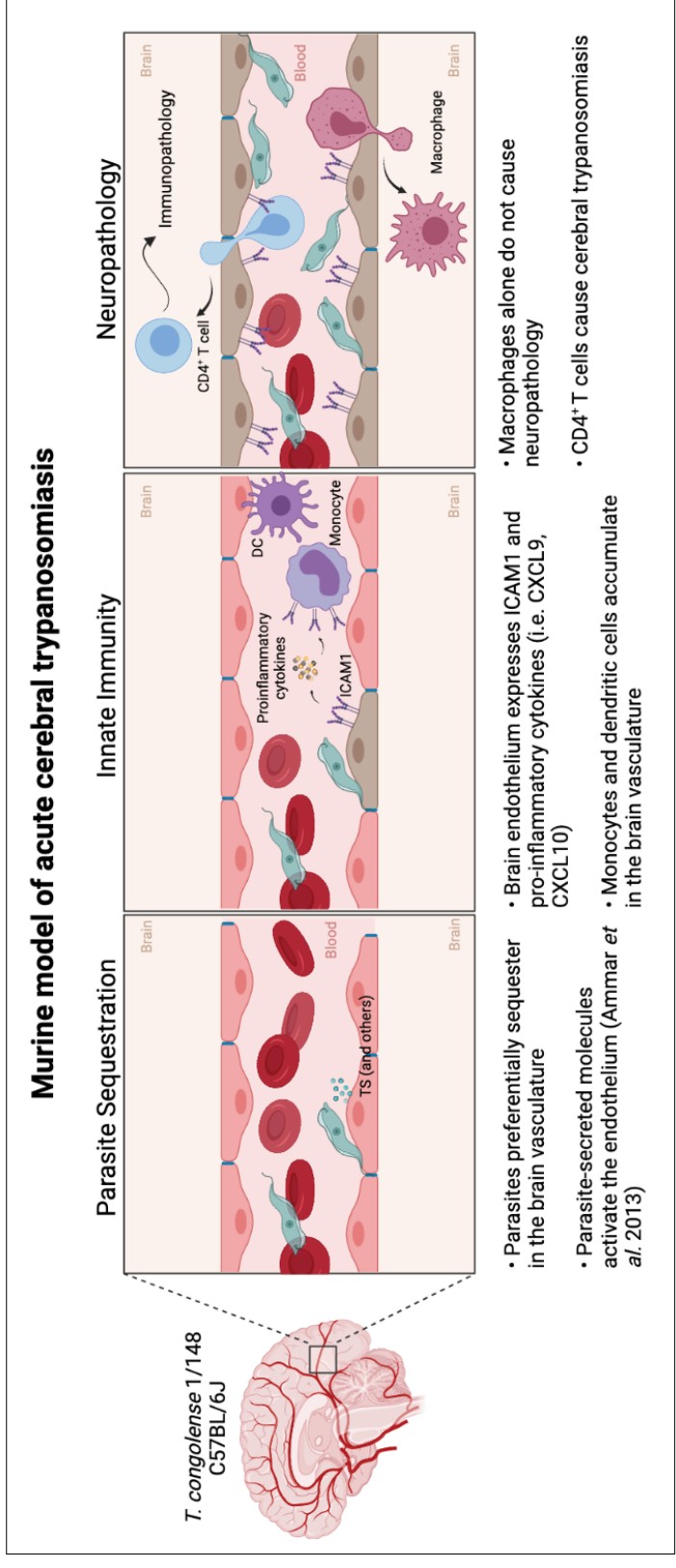

**Figure 8.** Murine model of cerebral animal African trypanosomiasis. Upon infection of C57BL/6 J mice with *T. congolense* 1/148 parasites cytoadhere to the brain vasculature ('Parasite Sequestration'). The physical damage caused by sequestration and the release of parasitic molecules, like trans-sialidases, activates the endothelium leading to an increase in expression and secretion of pro-inflammatory molecules (e.g. ICAM1, IFNγ, CXCL9,

*Figure 8 continued on next page*

*Figure 8 continued*

CXCL10), as well as the recruitment of innate cells (i.e. DC, monocytes) ('Innate Immunity'). Innate cells and activated endothelial cells recruit T cells to the brain vasculature and promote monocyte differentiation into inflammatory macrophages. Although both T cells and inflammatory macrophages can cross the blood-brain-barrier and infiltrate the brain parenchyma, only CD4+ T cells cause the neuropathology associated with acute disease ('Neuropathology'). Figure created with https://biorender.com/.

congolense savannah 1/148] (MBOI/NG/60/1–148) (*Young and Godfrey, 1983*) or $2 \times 10^4$ *T. congolense* savannah IL3000 (*Gibson, 2012*) parasites, under mild isoflurane anesthesia. Parasitemia was estimated daily by hemocytometry from tail venipuncture. Mice were sacrificed by anesthetic overdose (intravital and ex-vivo microscopy), cervical dislocation (flow cytometry) or $CO_2$ narcosis (all remaining experiments). Blood was collected by heart puncture and, when necessary, mice were perfused with 50 mL heparinized PBS. Tissues were dissected, washed in PBS and immediately imaged, snap frozen in liquid nitrogen, or fixed in 10% neutral-buffered formalin.

## Histology and immunohistochemistry

Organs were fixed in 10% neutral buffered formalin solution, embedded in paraffin and 3 µm sections were stained with hematoxylin and eosin (H&E). For immunohistochemistry, the tissue sections were pre-treated in a PT module (Fisher Scientific, New Hampshire, USA) at High-Ph, followed by incubation with the primary antibody (non-purified rabbit serumα-*T. brucei* histone 2 A (H2A) (generated against a recombinant protein) (kind gift of Christian Janzen), diluted 1:5000). EnVision Link horse-radish peroxidase/DAB visualization system (Dako, California, USA) was used, and sections were then counterstained with Harris hematoxylin and mounted. Tissue sections were examined by a pathologist, blinded to experimental groups, in a Leica DM2500 microscope coupled to a Leica MC170 HD microscope camera.

## Blood biochemistry analyses

Serum of mice infected with either *T. congolense* 1/148 or IL3000 was used for biochemistry analysis to determine blood levels of urea, creatinine and NGAL levels at the first peak of parasitemia. Samples were processed and analyzed by DNAtech (Portugal).

## Surgical procedures and intravital microscopy

For intravital microscopy, surgeries were separately performed by groups of organs, as previously described for the brain (*De Niz et al., 2019b*), the lungs and heart, the liver, pancreas, spleen, and kidneys (*De Niz et al., 2020*), and the adipose tissue (*De Niz et al., 2019a*). In summary, mice were anesthetized prior to surgery with a mixture of ketamine (120 mg/kg) and xylazine (16 mg/kg), by intra-peritoneal injection. Reflex responses were induced, and, once these responses were non-existent, mice were injected intravenously into the retro-orbital sinus with three markers: Hoechst 33,342 to label nucleic acids (stock diluted in dH2O at 100 mg/ml, injection of 40 µg/kg mouse), 70 kDa FITC-Dextran to label the intravascular space and provide contrast within the vessel walls (stock diluted in 1 x PBS at a concentration of 100 mg/ml, injection of 500 mg/kg mouse), and a fluorescently conjugated antibody against the pan-vascular marker CD31 (PECAM1) (BioLegend, used at 20 µg/mouse). For imaging, a temporary glass window (Merk rectangular cover glass, 100 mm x 60 mm) or a circular cover glass (12 mm) of 0.17 mm thickness was implanted in each organ. The windows were secured surgically using stitches, or immobilized using surgical glue. For imaging the heart and lungs, the windows were immobilized using a vacuum to prevent collapse of the thoracic cavity. For brain imaging, semi-closed or open cranial (with cranium removed, brain parenchyma fully exposed) windows were implanted, reaching either around 190 µm or up to 300 µm of depth into the tissue, respectively.

All intravital microscopy performed with the aim of identifying flowing and circulating parasites was performed in spinning disc microscopes. These included a Zeiss Cell Observer SD (Carl Zeiss Microimaging, equipped with a Yokogawa CSU-X1 confocal scanner, and an Evolve 512 EMCCD camera and a Hamamatsu ORCA-Flash 4.0 VS camera) or a 3i Marianas SDC (spinning disc confocal) microscope (Intelligent Imaging Innovations, equipped with a Yokogawa CSU-X1 confocal scanner and a Photometrics Evolve 512 EMCCD camera). Laser units 405, 488, and 647 were used to image Hoechst,

FITC-Dextran and AF67-CD31 respectively. Visualization was done using either an oil-immersion plan apochromat 63 x objective with 1.4 Numerical Aperture (NA) and 0.17 mm working distance (WD), or a 40 x LD C-Apochromat corrected, water immersion objective with 1.1 NA and 0.62 WD. Images were obtained for a total of 20 seconds, at an acquisition rate of 20 frames per second. For all acquisitions, the software used was either ZEN blue edition v.2.6., or 3i Slidebook reader v.6.0.22.

For the quantification of the duration of sequestration in the brain, longer imaging sessions were required. Therefore, *T. congolense*-infected mice were imaged at day 6 post-infection for a total of 12 hr, with images acquired every 30 mins. In this case, a non-invasive cranial window was implanted on the skull (i.e. no craniectomy performed, imaging is done through the skull), and anesthesia was injected every hour intraperitoneally, in addition to inhalable anesthesia (isoflurane) being supplied as required to ensure analgesic and anesthetic effect for the full imaging time. Sequestered parasites were defined as those that did not change positions relative to the immediately preceding time, in half-hourly periods.

## Ex-vivo microscopy
While intravital imaging in living mice mostly allows visualization of parasite-host interactions in the cerebral cortex, the meninges, and the olfactory bulb, we were interested in visualizing parasite dynamics throughout the entire brain. For this purpose, we followed the same procedures as previously described for vascular and nucleic acid labeling, and we then surgically exposed 10 different regions of the mouse brain, namely the olfactory bulb, the cerebral cortex, the septum, the hypothalamus, the thalamus, the hippocampus, the midbrain, the pons, the medulla, and the cerebellum. Imaging was performed within a humid chamber with 37 °C stable temperature and oxygen flow to delay loss of oxygenation and therefore alter parasite dynamics. Using these conditions, parasites retain their motility for an average of 1 hr, during which we quantified sequestered and flowing parasites in the vascular area of each region.

## Image analysis
### Quantification of sequestered and flowing parasites
In order to quantify sequestered and flowing parasites, we used as reference the 70 kDa FITC-Dextran and the Hoechst 33,342. On one hand, the FITC-Dextran enabled identification of the parasite bodies by the contrast generated in the vascular space. Meanwhile, the Hoechst 33,342 which stains nucleic acids, allowed clear labeling of the nucleus and kinetoplast of *T. congolense* parasites. Flowing parasites were defined as those displacing along the field of view throughout the time-lapse acquisition, together with the red blood cells (also identifiable by the contrast achieved by the FITC-Dextran within the vessel wall). Sequestered parasites were defined as those not displacing throughout the time-lapse acquisition. These parasites often displayed movement of the body and flagellum, but despite these local movement, the bodies remained attached at the same spot of the vessel wall. To enable quantifications by vascular area, we determined the total area of the field of view, and extracted the vascular area using the FITC-Dextran and AF647-CD31 markers. For each organ, we estimated vascular density ($D_{V\%}$) as $D_{V\%} = A_V /_{AT} \times 100$, where $A_V$ is the total vascular area imaged and $A_T$ is the sum of the areas of the field of view. Then, using the brain $D_{V\%}$ as a normalizer, we estimated the vascular density ratio for each organ and adjusted the quantity of sequestered and flowing parasites. These measurements were performed daily in *T. congolense* Tc1/148-infected C57BL/6 J mice, throughout 6 days of infection, at peaks of infection in *T. congolense* IL3000-infected C57BL/6 J mice, and at day 6 of infection in several antibody-treated C57BL/6 J mice as well as RAG2 KO mice. In addition to vascular area, we quantified vascular diameter of vessels in each field of view, to calculate number of flowing and sequestered parasites in four categories of vessel diameters (0–10 μm, 10–20 μm, 20–40 μm, and >40 μm). Vascular area and diameter measurements were calculated using Fiji software.

## Quantification of parasite region enabling sequestration
To evaluate and quantify parasite regions enabling sequestration, we used spinning disc confocal microscopy as previously described. Visualization was done using an oil-immersion plan apochromat 100 X objective with 1.4 NA and 0.17 WD. Throughout 5 min of time-lapse imaging at a rate of 20 frames per second, we determined whether sequestered parasites remained attached using the

mid-body, the parasite posterior, the flagellar tip, or variable. The latter was defined as several regions of the parasite involved in sequestration, with the attachment point shifting throughout the time lapse imaged.

## Leukocyte quantification by intravital microscopy

Intravital microscopy allowed us to observe an increase of nucleated, large cells in the vasculature of the brain in infected mice. This visualization is achieved by the contrast provided by the 70 kDa FITC-Dextran, and the nuclear labeling allowed by Hoechst 33,342. Although these combined dyes do not allow the definition of types and sub-types of leukocytes during infection, they allow segmentation and quantification of new cell populations compared to control uninfected mice. Having obtained information from intravital microscopy, we characterized immune cell populations in infected animals by flow cytometry, using specific antibodies as described below.

## Endothelial cell polysome immunoprecipitation and RNA sequencing

RiboTag.PDGFb-iCRE mice were bred in-house, at the iMM rodent facility, by crossing PDGFb-CreERT2 mice (Tg(Pdgfb-icre/ERT2, EGFP)1Frut) (*Claxton et al., 2008*) with RiboTag mice (RRID:IMSR_JAX:030201) (*Sanz et al., 2009*), in a n a C57BL/6 J background. Cre recombinase activity was induced in RiboTag.PDGFb-iCRE mice by daily i. p. injection of 10 µL/g mouse weight of 8 mg/mL of tamoxifen (Sigma Aldrich, T5648), diluted in 10% ethanol and 90% peanut oil (Sigma Aldrich, P2144), for 5 days. Mice were left to recover for 72 hr before infection with $2 \times 10^3$ *T. congolense* 1/148 parasites. Age and sex-matched mice were used as non-infected controls. Mice were euthanized at days 6 post-infection and perfused with 50 ml heparinized PBS. Brains were dissected, cut in half and homogenized in supplemented homogenization buffer (50 mM Tris, pH 7.5, 100 mM KCl, 12 mM $MgCl_2$, 1% Nonidet P-40, 1 mM DTT, 200 U/mL Promega RNasin, 1 mg/mL heparin, 100 µg/mL cycloheximide, Sigma protease inhibitor mixture) using a 2 ml dounce homogenizer and glass pestles. Ribosomes of the endothelial cells were immunoprecipitated according to *Sanz et al., 2009*. Briefly, homogenized brains were incubated with 4 µl of α-HA.11 Epitope Tag Antibody (BioLegend, UK) for 2 hr at 4 °C; polysomes bound to α-hemagglutinin antibody were immunoprecipitated using Pierce A/G magnetic beads (ThermoFisher Scientific, Spain) previously washed in homogenization buffer (50 mM Tris, pH 7.5, 100 mM KCl, 12 mM MgCl2, 1% Nonidet P-40). Beads bound to antibody-polysome conjugation were washed with high-salt buffer (50 mM Tris, pH 7.5, 300 mM KCl, 12 mM $MgCl_2$, 1% Nonidet P-40, 1 mM DTT, 100 µg/mL cycloheximide) and dissociated by chemical and mechanical lysis. RNA was extracted using the RNeasy Mini kit (Qiagen, UK).

RNA concentration and integrity were checked by fluorometry (Qubit DNA HS, Thermo Fisher Scientific) and parallel capillary electrophoresis (Fragment Analyzer, BioLabTech), respectively. cDNA libraries were prepared using the QuantSeq 3' mRNA-Seq Library Prep Kit FWD for Illumina (Lexogen, Austria), as per manufacturer instructions, and sequenced as 75 bp single-end reads on the NextSeq 550 platform (Illumina, USA). Sequencing reads are available from NCBI under BioProject accession number: PRJNA777781. Reads were aligned to the mouse genome GRCm39 (http://www.ensembl.org/Mus_musculus/Info/Index) using STAR (*Dobin et al., 2013*). The output from read alignment was processed with SAMtools (*Li et al., 2009*), and transcript abundances were estimated using stringtie (*Pertea et al., 2015*). Differential expression between input and immunoprecipitated samples was performed in R, using DESeq2 (*Love et al., 2014*). Log2 Fold change of 1 and p-value < 0.05 was considered significant. Gene Set Enrichment Analysis (GSEA) (*Subramanian et al., 2005*) was performed using the WEB-based GEne SeT AnaLysis Toolkit (WebGestalt) (*Wang et al., 2017*). Non-expressed genes were removed from the analysis and the remaining were ranked by differential expression Log2 fold change. Gene sets were defined based on gene ontology for biological function.

## Cytokine profiling

Blood was collected from mice either non-infected or infected with either *T. congolense* 1/148 or IL3000 at the first peak of parasitemia by cardiac puncture. Blood was allowed to clot for 30 min at room temperature and then centrifuged at 1000 x g for 10 min at 4 °C. Serum was collected from each sample and added to an equal volume of PBS (pH 7.4). Samples were immediately frozen at –80 °C, and shipped in dry ice to Eve Technologies (Canada), where a Mouse Cytokine Array / Chemokine Array 31-Plex was performed, in duplicate.

## Receptor expression by mean fluorescent intensity

To investigate the relative expression of endothelial receptors in different organs, we intravenously injected 20 μg per mouse, of fluorescently conjugated antibodies against ICAM1, ICAM2 (BioLegend, conjugated to AF647), and VCAM1 (Invitrogen, conjugated to FITC). These were administered intravenously, by retroorbital injection, into either non-infected mice or mice infected with *T. congolense* 1/148 or IL3000, at the first peak of parasitemia. We measured mean fluorescent intensities of at least 100 vessels per organ in 3 separate mice using an LSM 880 Zeiss microscope, and a 40 x oil objective (1.3 NA).

## Measurement of vascular leakage and vascular permeability by mean fluorescent intensity

To investigate vascular leakage in the brain, we used 70 kDa FITC-Dextran as reference, administered intravenously, by retroorbital injection, into either non-infected mice, mice infected with *T. congolense* 1/148 or IL3000 at first peak of parasitemia. We then quantified the MFI of FITC in the extravascular compartments, as a proxy dor vascular permeability. In healthy, uncompromised vasculature, 70 kDa FITC-Dextran remains intravascular, so the extravascular space shows minimal fluorescence. In contrast, when the vasculature is compromised, 70 kDa FITC-Dextran can leak out of the vasculature either. The pattern of 70 kDa FITC-Dextran can allow the identification of the type of vascular damage: increased vascular permeability results in uniformly increased fluorescence of the extravascular region most proximal to the vessel, microhemorrhaging results in strong, punctual accumulation of FITC-Dextran. To determine both, we calculated MFI within the 10 μm proximal to the nearest vessel, following background correction. Additionally, we assessed in a qualitative manner whether punctual hemorrhages were present. We measured MFI of at least 60 vessels in three separate mice using an LSM 880 Zeiss microscope, and a 40 x oil objective (1.3 NA). For qualitative assays, we classified vessels according to the following: no increased permeability and no hemorrhages; increased permeability but no hemorrhages; no increased permeability but hemorrhages present; and increased permeability and presence of hemorrhages.

## Blocking antibody binding assay

Mice were treated daily with 2 μg of α-ICAM1 antibody [CD54 monoclonal antibody (YN1/1.7.4), #16-0541-85, Invitrogen], or rat α-IgG2b kappa isotype control (eB149/10H5), #16-4031-81, Invitrogen, diluted in 200 μL PBS via retro-orbital intravenous injection, under isoflurane anesthesia. Twenty-four hours after the first antibody treatment, mice were infected with *T. congolense* 1/148 parasites as described above.

## rICAM1 static adhesion assay

To test the capacity of *T. congolense* 1/148 parasites to bind to ICAM1 in vitro, we performed a static adhesion assay. Plastic cell culture-grade 48-well plates (Corning, USA) were coated with 100 μL of protein A (#6,500B, BioVision) diluted at 20 μg/mL in PBS (pH 9.0) and incubated in a cell culture incubator for 1 hr at 37 °C, 5% $CO_2$. Plates were washed three times with PBS (pH 7.4) to remove unbound protein, blocked with 200 μL of 1% bovine serum albumin (BSA), and incubated at 4 °C overnight. Subsequently, plates were washed with PBS (pH 7.4) to remove unbound BSA,100 μL of purified recombinant mouse FC chimera protein [rICAM1 or rCD36 (#796-IC-050 and # 2519 CD-050, R&D Systems)], diluted at 25 μg/mL were spotted in a radial pattern onto the center of the wells, and incubated in a cell culture incubator for 3 hr at 37 °C, 5% $CO_2$. Plates were washed three times with PBS (pH 7.4) and 100 μL of BSA were added. Plates were incubated again for 30 min at 37 °C, 5% $CO_2$. Simultaneously, parasites were isolated from mouse by anion exchange chromatography (*Lanham and Godfrey, 1970*) and stained with 5 mM Vybrant CFDA SE Cell Tracer dye (#V12883, Invitrogen) diluted 1000 times in trypanosome dilution buffer (TDB) (5 mM KCl, 80 mM NaCl, 1 mM MgSO4, 20 mM Na2HPO4, 2 mM NaH2PO4, 20 mM glucose, pH 7.4), and incubated for 25 min at 34 °C, 5% $CO_2$. At the end of the incubation period, parasites were washed and resuspended in TDB, added to the previously washed pre-coated plates, and incubated for 1 hr at 34 °C, 5% $CO_2$. Plates were washed twice with PBS (pH 7.4) to remove unbound parasites. We added 100 μL of TDB to the washed plates, now containing only adhered parasites, and proceeded with live imaging on a Zeiss Cell Observer (Carl Zeiss Microimaging) with a 40 X water-immersion objective. We acquired 15 fields

of view per replicate well (3 replicates), per condition (BSA alone, rICAM1, rCD36) using bright-field light and green laser. We used Fiji software to count adhered parasite per field of view and total area imaged.

## SHIRPA test

We performed the primary screen of the previously described SHIRPA protocol (*Rogers et al., 1997*), consisting on the behavioral observation profiling of mice. The SHIRPA test allows assessment of muscle and lower motor neuron function, spinocerebellar function, sensory function, neuropsychiatric function, and autonomic function. The parameters assessed and their respective scores were: Locomotion: 0 – Completely flattened, 1 – Lying on the side, 2 – Lying upright, 3 – Sitting up, 4 – Standing on hindlimbs (Rearing) (Normal), 5 – Repeated vertical leaping; Posture: 0 – Normal, 1 – Hunched, 2 – No posture; Exploration/Spontaneous activity: 0 – Inactive, Resting, 1 – Active (Normal), 2 – Excessively Active; Movement Type (and tail posture): 0 – Normal movement, tail up, 1 – Walking well, but tail lagging behind, 2 – Dragging the lower body, 3 – Inactive; Breath Rate: N(0)– Normal, L(1) – Labored, S(2) – Shallow, R(3) – Retching, D(4) – Dyspneic, G(5) – Gasping; Palpebral closure: 0 – Closed, 1 – Semi-opened, 2 – Opened/Normal 3 – Extremely opened; Touch Escape: 0 – None, 1 – Slow escape, 2 – Moderate/rapid escape (Normal), 3 – Rapid escape; Coat Appearance: N(0) – Normal grooming, normal density, P(1) – Moderate piloerection, PP(2) – Extreme piloerection; Positional Passivity: 0 – No struggle, 1 – Struggle, open limbs, 2 – Struggle, grasping limbs, closed paws, 3 – struggle, trunk curl; Grip Strength: Time (in seconds) that mice can hold upside down on a grid (Grid Test, maximum of 1 min). All tests were performed in a new, clean cage, so that the mice would not be disturbed by cage mates.

## Immune cell isolation, staining and flow cytometry

Infected and non-infected mice were anaesthetized with isoflurane and received 3 µg of α-APC-CD45 intravenously, by retroorbital injection. Antibodies were allowed to circulate for a maximum of three minutes, mice were euthanized by cervical dislocation, 50–150 µL of blood were collected by cardiac puncture, added to 2 mL ACK lysis buffer (155 mM Ammonium Chloride, 10 mM Potassium Bicarbonate, 0.1 mM EDTA), left to incubate at room temperature for 5 min for red blood cell lysis, and centrifuged for 5 min at 550 x g, 4 °C. The supernatant was discarded the procedure repeated. At the end, the cell pellet was gently resuspended in RPMI 1640 medium (#11875093, Gibco) supplemented with 10% FBS (#10270106, Gibco). Brains were dissected, added to 5 mL supplemented RPMI 1640 medium, cut in small pieces, and incubated with 100 µg/mL DNAse and 1.5 mg/mL collagenase D for 30 min at 37 °C with periodic agitation. After the incubation, an additional 15 ml of medium were added, the organs were forced to pass through a 70 µm cell strainer, and centrifuged at 550 x g for 5 min at 4 °C. Resulting pellet was resuspended in 3 mL of 40% percoll and passed to a 15 mL tube, after which 2 mL of 70% percoll were slowly added to the bottom. Samples were centrifuged for 30 min at 2400 rpm with no acceleration or brake. The interphase was carefully collected with a pipette, added to 5 ml of supplemented RPMI 1640 medium, and centrifuged at 550 x g for 5 min at 4 °C. The supernatant was discarded by inversion and the pellet resuspended in 2 ml of supplemented RPMI 1640 medium. Isolated blood and brain immune cells were counted on a hemocytometer, and between $2.5 \times 10^5$ and $2 \times 10^6$ cells per sample stained with the following conjugated mouse antibodies: FITC-Ly6C (#128005, Biolegend), PE/CD11c (#117307, Biolegend), PerCP-Cy5.5-CD8 (#100733, Biolegend), PE/Cy7-F4/80 (#123113, Biolegend), PE/Dazzle-CD45 (#103145, Biolegend), Alexa Fluor 700-CD11b (#101222, Biolegend), APC/Cy7-CD4 (#100413, Biolegend), BV421-ICAM1 (#565987, BD Biosciences), BV605-CD19 (#563148, BD Biosciences), BV711-CD3 (#740739, BD Biosciences), BV785-MHCII (#107645, BD Biosciences), and CD16/CD32 FcR-blocking reagent (#553141, BD Biosciences). Live and dead cells were separated using BV510-Zombie Aqua fixable viability dye (4231010, Biolegend). Acquisition of mean fluorescence intensities was performed on the BD LSRFortessa X-20 Cell Analyzer equipped with a Violet (406 nm, 100 mW), Blue (adjustable 488 nm, 80 mW; maximum output 100 mW), Green (532 nm, 150 mW) and Red (642 nm, 40 mW) lasers, then analyzed with FlowJo v10 (TreeStar Technologies).

## Monocyte/macrophage depletion

Circulating monocytes and inflammatory macrophages were depleted by intravenous retro-orbital injection of 1 mg clodronate liposomes (Liposoma BV, The Netherlands) 24 hr prior to infection. Two control groups were included, one receiving PBS liposomes, and the other receiving PBS only. Treatment was repeated 3 days post-infection to maintain depletion.

## Adoptive transfer of lymphoid cells

Spleens and lymph nodes of 3 C57BL/6 J mice were dissected into 2 mL of RPMI 1640 medium (#11875093, Gibco) supplemented with 10% FBS (#10270106, Gibco) and passed through a 70-μm cell strainer. To this homogenate, 2 mL ACK lysis buffer (155 mM Ammonium Chloride, 10 mM Potassium Bicarbonate, 0.1 mM EDTA) was added, left to incubate at room temperature for 5 min for red blood cell lysis, then centrifuged for 5 min at 550 x g, 4 °C. The supernatant was discarded and the cell pellet was gently resuspended in RPMI 1640 medium (#11875093, Gibco) supplemented with 10% FBS (#10270106, Gibco). Cells were stained with the following conjugated mouse antibodies: PE-CD8 (#100707, Biolegend), APC-CD4 (#100411, Biolegend), Alexa Fluor 700-CD11b (#101222, Biolegend), APC/Cy7-NK1.1 (#108723, Biolegend), BV605-CD19 (#563148, BD Biosciences), and CD16/CD32 FcR-blocking reagent (#553141, BD Biosciences). Live and dead cells were separated using BV510-Zombie Aqua fixable viability dye (4231010, Biolegend). $CD4^+$, $CD8^+$ and $CD19^+$ cells were sorted and acquired on the BD Aria III into 5 mL tubes containing RPMI 1640 medium supplemented with 10% FBS. Recovered cells were centrifuged for 5 min at 550 x g, 4 °C, and resuspended in enough volume of RPMI 1640 medium supplemented with 10% FBS. We injected either $5 \times 10^6$ $CD19^+$ B cells, $2.6 \times 10^6$ $CD4^+$ T cells, $2 \times 10^6$ $CD8^+$ T cells, or PBS in a volume of 200 μl into naïve RAG2 KO mice (3 per experimental group) by retroorbital injection, under mild isoflurane anesthesia.

## Acknowledgements

We thank Dr Álvaro Acosta-Serrano, at the Liverpool School of Tropical Medicine, and Dr Loïc Rivière for providing *T. congolense* 1/148 and IL3000 parasite lines, respectively. We thank the Silva-Santos lab for the providing the RAG2 KO mice. We are grateful to Dr Margarida Vigário for careful reading of the manuscript. We thank the Rodent, Bioimaging, and Flow Cytometry facilities and the Comparative Pathology Unit (including previous members Pedro Ruivo, DMV and Dr Tânia Carvalho, DMV) at iMM. This work was supported by European Union's Horizon 2020 research and innovation program through a Marie Skłodowska-Curie Individual Standard European Fellowship to S.S.P., under grant agreement no. 839960, and from the European Research Council (ERC) (FatTryp, 771714) to L.M.F. M.D.N. was funded by Human Frontiers LT000047/2019 L (HFSP) and EMBO (ALTF 1048–2016). L.M.F., K.S., and C.A.F. are Investigators CEEC of the Fundação para a Ciência e a Tecnologia (CEECIND/03322/2018, CEECIND/00697/2018, CEECIND/04251/2017, respectively). C.A.F. was supported by a European Research Council starting grant (679368), the Fondation Leducq (17CVD03), and the Fundação para a Ciência e a Tecnologia (grants IF/00412/2012, EXPL/BEX- BCM/2258/2013, PRECISE-LISBOA-01–0145-FEDER-016394, PTDC/MED-PAT/31639/2017, PTDC/BIA-CEL/32180/2017).

## Additional information

### Competing interests

The other authors declare that no competing interests exist.

### Funding

| Funder | Grant reference number | Author |
| --- | --- | --- |
| European Research Council | 771714 | Luisa M Figueiredo |
| Human Frontier Science Program | LT000047/2019-L | Mariana De Niz |

| Funder | Grant reference number | Author |
| --- | --- | --- |
| European Molecular Biology Organization | ALTF 1048-2016 | Mariana De Niz |
| Horizon 2020 | Marie Skłodowska-Curie Actions (839960) | Sara Silva Pereira |
| Fundação para a Ciência e a Tecnologia | CEECIND/03322/2018 | Luisa M Figueiredo |
| Fondation Leducq | 17CVD03 | Cláudio A Franco |
| Fundação para a Ciência e a Tecnologia | IF/00412/2012 | Cláudio A Franco |
| European Research Council | 679368 | Cláudio A Franco |
| Fundação para a Ciência e a Tecnologia | CEECIND/04251/2017 | Cláudio A Franco |
| Fundação para a Ciência e a Tecnologia | CEECIND/00697/2018 | Karine Serre |
| Fundação para a Ciência e a Tecnologia | PTDC/MED-PAT/31639/2017 | Cláudio A Franco |
| Fundação para a Ciência e a Tecnologia | PRECISE-LISBOA-01-0145-FEDER-016394 | Cláudio A Franco |
| Fundação para a Ciência e a Tecnologia | EXPL/BEX-BCM/2258/2013 | Cláudio A Franco |

The funders had no role in study design, data collection and interpretation, or the decision to submit the work for publication.

## Author contributions

Sara Silva Pereira, Conceptualization, Formal analysis, Investigation, Methodology, Validation, Writing – original draft; Mariana De Niz, Formal analysis, Investigation, Methodology, Validation, Writing – original draft; Karine Serre, Conceptualization, Formal analysis, Writing – review and editing; Marie Ouarné, Investigation; Joana E Coelho, Methodology; Cláudio A Franco, Conceptualization, Funding acquisition, Resources; Luisa M Figueiredo, Conceptualization, Funding acquisition, Resources, Supervision, Writing – original draft, Writing – review and editing

## Author ORCIDs

Sara Silva Pereira http://orcid.org/0000-0002-6590-6626
Mariana De Niz http://orcid.org/0000-0001-6987-6789
Karine Serre http://orcid.org/0000-0001-9152-4739
Marie Ouarné http://orcid.org/0000-0003-4724-4363
Joana E Coelho http://orcid.org/0000-0002-3964-6197
Cláudio A Franco http://orcid.org/0000-0002-2861-3883
Luisa M Figueiredo http://orcid.org/0000-0002-5752-6586

## Ethics

This study was conducted in accordance with EU regulations and ethical approval was obtained from the Animal Ethics Committee of Instituto de Medicina Molecular (AWB_2016_07_LF_Tropism). All surgeries were performed under anaesthesia, and every effort was made to minimize suffering.

## Decision letter and Author response

Decision letter https://doi.org/10.7554/eLife.77440.sa1
Author response https://doi.org/10.7554/eLife.77440.sa2

# Additional files

## Supplementary files

• Supplementary file 1. Sequencing statistics and transcripts detected from ribosome profiling of the

brain endothelial cells at day 6 post-infection with *T. congolense* 1/148.

• Supplementary file 2. Input gene list, gene set results, and annotation of the Gene Set Enrichment Analysis (GSEA) of transcripts upregulated in the brain endothelial cells upon infection with *T. congolense* 1/148.

• Transparent reporting form

• Source data 1. Source data of main figures.

• Source data 2. Source data of figure supplements.

## Data availability

Sequencing reads are available from NCBI under BioProject accession number: PRJNA777781.

The following dataset was generated:

| Author(s) | Year | Dataset title | Dataset URL | Database and Identifier |
|---|---|---|---|---|
| Silva Pereira S, De Niz M, Serre M, Ouarné M, Coclho JE, Franco CA | 2021 | Ribosome profiling of *M. musculus* brain during T. congolense infection | https://www.ncbi.nlm.nih.gov/bioproject/?term=PRJNA777781 | NCBI BioProject, PRJNA777781 |

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
