## [Editor Report]

*Trypanosoma congolense* is an important animal trypanosome that exhibits significant biological differences to the better studied *Trypanosoma brucei*. In this study, the authors describe a novel mouse model of cerebral trypanosomiasis, based on the 1/148 strain of *T. congolense*. Using elegant intravital imaging, the authors show that parasites sequester in the vessels of various organs, especially in the brain, causing deleterious T cell responses and inflammation.

---

## [Decision Letter]

**Decision letter after peer review:**

[Editors’ note: the authors submitted for reconsideration following the decision after peer review. What follows is the decision letter after the first round of review.]

Thank you for submitting the paper "Immunopathology and *Trypanosoma congolense* parasite sequestration cause severe cerebral trypanosomiasis" for consideration by *eLife*. Your article has been reviewed by 3 peer reviewers, one of whom is a member of our Board of Reviewing Editors, and the evaluation has been overseen by a Senior Editor. The reviewers have opted to remain anonymous.

Comments to the Authors:

We are sorry to say that, after consultation with the reviewers, we have decided that this work will not be considered further for publication by *eLife*.

Specifically, while the reviewers agree that the work is of high quality, they also raised important questions regarding the completeness and relevance of the study. In particular, a causative link between parasite sequestration and immunopathology and disease severity is not firmly established, and the relevance of the model as regards to animal trypanosomiasis in the field is uncertain.

*Reviewer #1 (Recommendations for the authors):*

This is an interesting study describing a novel model of cerebral trypanosomiasis based on *T. congolense* 1/148 strain infection in C57BL/6 mice. The authors show that the 1/148 strain of *T. congolense* (but not the IL3000 strain) provokes a lethal acute disease in C57BL/6J mice, associated with neurological symptoms and brain histological lesions. Using elegant intravital imaging, the authors show that parasites sequester in the vessels of various organs, especially in the brain. Analysis of the transcriptome of brain endothelial cells revealed a pro-inflammatory response in mice infected with 1/148 as compared to uninfected mice. The authors further show that ICAM1 is overexpressed in the brain of infected mice and that blockage of ICAM1 using antibodies reduces disease severity and parasite sequestration in the brain. They also analyzed cellular responses and document that the development of cerebral trypanosomiasis is associated with the recruitment of ICAM1-expressing myeloid cells to the brain vasculature. Finally, infections performed in RAG2 KO mice revealed that cerebral trypanosomiasis is prevented in the absence of T cells. Altogether the data suggest that cerebral trypanosomiasis results from a combination of parasite sequestration and pathogenic inflammatory responses. This study reveals a remarkable similarity between cerebral trypanosomiasis and experimental cerebral malaria, where parasite sequestration and T cell responses are responsible for a lethal neuroinflammatory disease in mice.

Strengths:

– This study addresses an important problem as animal African trypanosomiasis is a neglected yet devastating disease in endemic countries, for which animal models are needed. This study provides a novel and useful mouse model to study cerebral trypanosomiasis.

– The manuscript is very well written and the work is well conducted. It includes a thorough characterization of parasite behavior in vivo, based on elegant intravital imaging, as well as in-depth cellular analyses.

– The data convincingly document that 1/148 disease in mice is associated with brain lesions and neuroinflammation combined with parasite sequestration, and highlight the role of ICAM1 and T cells.

Limitations:

– The IL3000 strain shows some levels of sequestration in the brain, yet is not associated with acute lethality. This raises the question of the role of sequestration in disease onset.

– The contribution of ICAM1 expressed by endothelial versus blood cells is unclear.

– The role of T cells in the development of cerebral trypanosomiasis is not directly addressed.

Both parasite sequestration and inflammatory immune responses seem to contribute to the disease. The authors should take advantage of the IL3000 strain to dissect the pathogenic mechanisms further. In particular, although the IL3000 does not provoke a lethal acute infection, it is associated with significant parasite sequestration in the brain (figure 2). In this context, it is important to include this strain (in addition to uninfected mice) in some of the experiments addressing endothelial cell responses, ICAM1 expression and cellular responses.

The data with the RAG2 KO mice suggest an important role of T cells. This could be addressed directly, for example through antibody-mediated depletion of CD4 or CD8 T cells, or via adoptive transfer experiments.

The imaging work is beautiful, however, some additional technical explanations would be useful in the text. For example, the Hoechst dye stains all the cells, not only the parasite. In many figures (for example 2B, 5D and 7A, or movie 1) it is difficult to distinguish if cells are leukocytes or parasites. Maybe the authors could show higher magnification insets to illustrate how they could distinguish host versus parasite cells. Also, the authors could provide examples of images where one can discriminate cytoadhered versus flowing parasites (in movie 4, there seems to be a flowing cell, but it could be a leukocyte).

Previous studies in rats and mice have shown that *T. brucei* infection is associated with parasite and T cells extravasation. Did the authors observe similar phenomena with *T. congolense*? Could they document a rupture of the blood brain barrier integrity? Did they observe dextran leakage? Are they sure that the injected anti-CD45 antibody does not diffuse in extravascular compartments due to alteration of the blood brain barrier?

Is there a way to disentangle the contribution of endothelial versus leukocyte ICAM1? In Figure 4F, it seems that ICAM1 increased signal comes from intravascular cells (presumably leukocytes), not endothelial cells.

In figure 5D the vessels imaged after injection of ICAM1 antibody look very different (smaller) than with the control anti-IgG2. The authors should show similar vessels to allow comparisons.

In the in vitro binding experiments (Figure 5 F-G) the authors used parasites isolated from mouse blood. Is it possible that cytoadhered parasites represent only a fraction of the total parasite population, which would be missed by sampling only circulating parasites?

Figure 4C is too small, difficult to read.

In videos S2, S3 and S4 the time scale should be provided. In video S2, all cells look non motile, as if there was no flow, why? In video S4, only parasites are visible in the vessel lumen, no other cells (leukocytes, erythrocytes), why?

*Reviewer #2 (Recommendations for the authors):Trypanosoma congolense* is an important animal trypanosome that exhibits significant biological differences to the better studied Trypanosoma brucei. Of note, the parasites sequester by binding within the vasculature and organs, one pathological consequence of which can be cerebral trypanosomiasis. In this manuscript the authors identify a strain of Trypanosoma congolense, 1/148, that exhibits extreme virulence and cerebral pathology, resulting in mouse death within a few days. This differs from another well-studied strain, IL3000, that generates sustained infections without cerebral pathology.The study initially characterizes the pathology and distribution of the respective *T congolense* parasite lines, using comprehensive imaging and histology to identify the parasites and their sequestration in different tissues. The appearance of large cerebral lesions is characteristic of *T congolense* 1/148 although the extent to which these are certain to be the cause of the ultimate death of the animal is perhaps overstated. It was also unclear how representative this acute and drastic cerebral involvement is of infections in the field. In the text, there is also some lack of clarity regarding the distinction between sequestration and cytoadherence. The authors take care to distinguish between these two phenomena (line 142-143) but then apparently use the phrase interchangeably in describing some of their observed parasite distributions on Figure 3 and line 220 forwards. It would be helpful to be more rigorous in the distinction if indeed it is important to understand the respective phenotypes of the parasite lines. Nonetheless, the descriptions of the parasite distribution appear high quality.

After characterization of the distribution of the parasites in various tissues, the immune responses to the infection are analyzed by gene expression profiling and then the serum analysis of various cytokines. The elevation of ICAM1 is also associated with the infection although the overall strength of upregulation appears modest. Regardless, the effect of blocking ICAM1 is apparently to reduce the pathology of the infection with 1/148 and associated immune responses are also reduced, providing evidence that the ICAM1 response is disease-relevant. However, in these studies, and more broadly in the analysis of various immune responses, I would have valued a comparison of the IL3000 (non-cerebral) and 1.148 (cerebral) responses rather than infection versus non infection to gain better reassurance of the specificity of responses for the pathological measure under assessment.

I lack detailed expertise, but it seemed the analysis of several immune responses to 1/148 infection were limited to rather blunt tools in some cases. Where more targeted knock out murine strains are available, chemical inhibition of macrophage using clodronate liposomes was used. Similarly, RAG2 KO cells were used to explore general T cell contributions but this also prevents antibody responses (notwithstanding the acute infection being analyzed where B cells may be less important). Again, more targeted interventions might be possible.

It was also intriguing that chronic infections could be sustained in RAG2 KO mice in the absence of antibody control. What do the authors think is happening here? Are the antigenic variation and periodic killing of antigenic variants that characterizes African trypanosomes less important for long term infections than widely considered?

In summary, I consider this a valuable model for exploring cerebral trypanosomiasis caused by Trypanosoma congolense. The very acute nature of the infection/pathology is striking and it would be useful to know if this is typical of cerebral trypanosomiasis or another facet of 1/148 that could contribute to some of the responses measured here but less applicable in disease-relevant isolates. The immunological studies are also somewhat preliminary and potentially correlative but reviewers with expertise in this specific area would likely have a better insight into the extent to which causative effects are being studied.

I would be reassured by a more comprehensive comparison between 1/148 and IL3000 to allow a better comparison between cerebral and non-cerebral trypanosomiasis rather than comparisons between infected and uninfected mice. This relates to gene expression studies and the analysis of specific immunological responses.

I would also prefer to see more specific targeting of immune phenotypes using specific knockout mouse strains where available to improve the resolution of the study compared to ablation of macrophages using liposomal killing, or quite generic RAG2 KO lines, for example.

*Reviewer #3 (Recommendations for the authors):*

This is the first time a detailed model for *T. congolense* brain pathology is being reported. The results are interesting, and a set of very well executed detailed experiments show the involvement of the ICAM1-CXCL9/10-IFNgamma- T cell axis. This is an interesting observation as it mirrors findings of the *T. brucei* models for human trypanosomiasis, with the big difference that the latter involves infiltration of the parasite into the brain parenchyma, while *T. congolense* remains in the microvasculature.

Despite the strengths of the experimental procedures, and the vast amount of data provided, it is not clear to which extent this unique mouse model is relevant for the in vivo field situation of *T. congolense* infections in cattle. In the case of the latter life-threatening pathology relates mostly to anaemia and a metabolic wasting disorder, rather than neuropathology. In that aspect, while interesting from a fundamental point of view, the paper might not shed new light on the true nature of animal trypanosomosis. In this context, attention should be given to the references used to support the potential role of neuropathology in AT: Tuntasuva et al. 1997 is a T. evansi paper (a very different parasite closely related to T. brucei, and not T. congolense), and so are the Rodriguez 2009 paper and the Busher 2019 paper, dealing with equine trypanosomosis caused by *T. evansi* and *T. equiperdum*. Finally, the Harrus 1995 paper is indeed a *T. congolense* case description of 2 dogs but is not a reference to livestock disease. Hence, none of the references support the idea that cattle or livestock trypanosomosis (the true problem of *T. congolense*) is hallmarked by neuropathology. For clarity towards the reading public, this section of the introduction should provide a more accurate description.

The second fundamental concern that requires a detailed discussion/consideration is the fact that only the rather unique acute 1/148 model studied in this paper seems to result in marked brain pathology in ice. In contrast, the well-described IL3000 model does not exhibit this feature. Taken than in the field *T. congolense* is a chronic infection, once again the question arises as to how relevant the observations of the acute rapid killing mode are.

Finally, the authors speculate on why they assume brain pathology is the cause of death of the infected mice, they acknowledge the occurrence of acute kidney disease at the time of death as ell. Hence, it is hard to see how the latter can be ignored as a cause of death as kidney failure has been described in other experimental trypanosome models and can result in t in sudden death within hours.

Overall, this is a very impressive work and it shows how modern techniques in experimental immunology can be combined to get a comprehensive vision on the development of infection-associated multi-organ pathology. It remains to be seen if this impressive knowledge truly relates to the disease of livestock trypanosomosis.

This is an impressive paper and the execution of the work is done at a level where this reviewer feels it would be almost unfair to request any additional experiments. The only concern is that the results might 'only' explain one unique condition of acute mouse T. congolese trypanosomosis. There are many models for *T. congolense* described in literature, and virtually all of them are characterized by multiple parasitemia peaks. And also in the field, *T. congolense* livestock infections are chronic wasting diseases, not acute neuropthological diseases with a quick deadly outcome. That would be a typical problem for *T. equiperdum*, but not T. congolense. The open discussion about this issue is missing from the paper. This could be improved, as the data itself is valuable by istelf, even if they do not represent the general pathology of livestock *T. congolense* AT.

[Editors’ note: further revisions were suggested prior to acceptance, as described below.]

Thank you for resubmitting your work entitled "Immunopathology and *Trypanosoma congolense* parasite sequestration cause acute cerebral trypanosomiasis" for further consideration by *eLife*. Your revised article has been evaluated by Dominique Soldati-Favre (Senior Editor) and a Reviewing Editor.

The manuscript has been improved but there are some remaining issues that need to be addressed, as outlined below:

– We kindly ask you to tone down the stated general biological importance of the acute model and the risk the acute pathology generates in wider geographical areas (see comments of reviewer #2).

*Reviewer #1 (Recommendations for the authors):*

The authors made a great job and have addressed all previous comments. The comparison between 1/148 and IL3000 strains now illustrates more convincingly that differences in the immune responses, especially CD4+ T cells, are central to the acute cerebral pathology.

*Reviewer #2 (Recommendations for the authors):*

The authors have made very substantial changes to the presented manuscript- particularly by increasing the comparative analysis between *T. congolense* isolates causing chronic (IL3000) versus acute (1/148) disease with cerebral involvement. These comparisons have been informative and addressed my main experimental comments from the last submission, further revealing the contribution of CD4^+^ T cells to the pathological phenotype. I am satisfied with the changes made in the experimental work given the constraints and what is possible (e.g., the RNAseq experiments could not be extended), though a reviewer with greater immunological expertise will provide better insight.

I remain a little hesitant on the relevance and value of the model with respect to *T. congolense* infection and pathology however. The authors' text previously mis-stated the importance of cerebral disease in livestock infections for *T. congolense*. In the new introduction they have revised this to reflect the evidence that other species (dogs, horses, goats, productive Bos taurus breeds) can suffer acute disease, and invoke the 1/148 model as a tool to address this. However, it seems the acute pathology is a consequence of the host and not the parasite in these instances, and so an alternative animal model rather than an additional parasite model such as 1/148 would be the more useful tool.

The risk of tsetse range expansion to threaten livestock susceptible to acute *T. congolense* infection also seems overstated. Unlike many hematophagous flies- e.g., mosquitos or sandflies- the biology of tsetse limits their capacity to significantly increase their range and evidence to date suggests that climate change has rather had the opposite effect. For other African trypanosomes (*T. vivax* and *T. brucei*) spread outside the tsetse belt has been possible by mechanical transmission but I am not aware of evidence supporting persistent and sustained spread of *T congolense* by this mechanism (despite there being one report of inefficient spread by tabanids). Hence the potential risk to susceptible breeds in other geographical settings may be very limited.

Overall, therefore, I applaud the additional experimental data provided which usefully enhances the paper from its original form and will be of broad interest, I think. I do retain some doubts over the general value of the 1/148 model, however and some of the wider biological implications of the work which potentially remain limited.

*Reviewer #3 (Recommendations for the authors):*

This reviewer would like to thank the authors for addressing the concerns posted in the previous round of reviewing in a very thorough manner. Addressing the kidney and the CD4 T cell issue was done by a convincing experimental approach, and the discussion on whether or not cerebral complications are a major contributor to *T. congolense* livestock death is now addressed by a more balanced phrasing as well as new and recent references. Hence, all major comments put forward by the reviewer in the past have been addressed and the paper has become an even more impressive piece of work. It will probably inspire other researchers in the field of AT to pay more attention to this pathology aspect.

---

## [Author Response]

[Editors’ note: The authors appealed the original decision. What follows is the authors’ response to the first round of review.]

Reviewer #1 (Recommendations for the authors):This is an interesting study describing a novel model of cerebral trypanosomiasis based on T. congolense 1/148 strain infection in C57BL/6 mice. The authors show that the 1/148 strain of T. congolense (but not the IL3000 strain) provokes a lethal acute disease in C57BL/6J mice, associated with neurological symptoms and brain histological lesions. Using elegant intravital imaging, the authors show that parasites sequester in the vessels of various organs, especially in the brain. Analysis of the transcriptome of brain endothelial cells revealed a pro-inflammatory response in mice infected with 1/148 as compared to uninfected mice. The authors further show that ICAM1 is overexpressed in the brain of infected mice and that blockage of ICAM1 using antibodies reduces disease severity and parasite sequestration in the brain. They also analyzed cellular responses and document that the development of cerebral trypanosomiasis is associated with the recruitment of ICAM1-expressing myeloid cells to the brain vasculature. Finally, infections performed in RAG2 KO mice revealed that cerebral trypanosomiasis is prevented in the absence of T cells. Altogether the data suggest that cerebral trypanosomiasis results from a combination of parasite sequestration and pathogenic inflammatory responses. This study reveals a remarkable similarity between cerebral trypanosomiasis and experimental cerebral malaria, where parasite sequestration and T cell responses are responsible for a lethal neuroinflammatory disease in mice.Strengths:– This study addresses an important problem as animal African trypanosomiasis is a neglected yet devastating disease in endemic countries, for which animal models are needed. This study provides a novel and useful mouse model to study cerebral trypanosomiasis.– The manuscript is very well written and the work is well conducted. It includes a thorough characterization of parasite behavior in vivo, based on elegant intravital imaging, as well as in-depth cellular analyses.– The data convincingly document that 1/148 disease in mice is associated with brain lesions and neuroinflammation combined with parasite sequestration, and highlight the role of ICAM1 and T cells.Limitations:– The IL3000 strain shows some levels of sequestration in the brain, yet is not associated with acute lethality. This raises the question of the role of sequestration in disease onset.– The contribution of ICAM1 expressed by endothelial versus blood cells is unclear.– The role of T cells in the development of cerebral trypanosomiasis is not directly addressed.Both parasite sequestration and inflammatory immune responses seem to contribute to the disease. The authors should take advantage of the IL3000 strain to dissect the pathogenic mechanisms further. In particular, although the IL3000 does not provoke a lethal acute infection, it is associated with significant parasite sequestration in the brain (figure 2). In this context, it is important to include this strain (in addition to uninfected mice) in some of the experiments addressing endothelial cell responses, ICAM1 expression and cellular responses.

We have done immunofluorescence analyses (IFA) of endothelial ICAM1, VCAM1, and ICAM2 expression in IL3000-infected mice, as well as serum cytokine profiling, and immunophenotyping to describe cellular responses upon IL3000 infection.

– IFA showed that ICAM1 expression was increased in both 1/148- and IL3000-infected mice. ICAM2 was only increased in IL3000-infected mice and VCAM1 was not increased in any of the models. This is presented in Figure 3E.

– Serum cytokine profiling showed that CXCL10 and CXCL9 are also increased in IL3000 infections, although to lower levels than 1/148. This is presented in Figure 4A.

– We then performed immunophenotying by flow cytometry to assess the changes of leukocyte subsets upon each infection model. We observed that, in contrast to infection by IL3000, the infection by 1/148 showed increased levels of ICAM1-expressing intravascular monocytes/macrophages and higher levels of extravascular T helper cells (CD4^+^). Furthermore, we observed that in IL3000 infections, there is an increase in ICAM1-expressing B cells in the brain parenchyma. This is now presented in Figure 4.

These results suggest that ICAM1 is important in both acute and chronic models of disease, but for different immunological responses (humoral in IL3000 and T cells in 1/148), which is consistent with the fact that in the chronic IL3000 infection, parasitaemia waves resulting from antibody clearance of parasites are observed.

The data with the RAG2 KO mice suggest an important role of T cells. This could be addressed directly, for example through antibody-mediated depletion of CD4 or CD8 T cells, or via adoptive transfer experiments.

We thank the reviewer for these suggestions. We have performed an adoptive transfer experiment to directly address the role of T cells and to decipher whether CD4^+^ or CD8^+^ T cells could drive neuropathology and acute disease. We found that transfer of CD4^+^ T cells alone into naïve Rag2 KO mice was sufficient to induce acute disease. This agrees with immunophenotyping results, where we observed that brain extravascular CD4^+^ T cells were significantly increased in 1/148 infection. We have included a new figure presenting this data (Figure 7) and updated the model presented in Figure 8.

The imaging work is beautiful, however, some additional technical explanations would be useful in the text. For example, the Hoechst dye stains all the cells, not only the parasite. In many figures (for example 2B, 5D and 7A, or movie 1) it is difficult to distinguish if cells are leukocytes or parasites. Maybe the authors could show higher magnification insets to illustrate how they could distinguish host versus parasite cells. Also, the authors could provide examples of images where one can discriminate cytoadhered versus flowing parasites (in movie 4, there seems to be a flowing cell, but it could be a leukocyte).

We thank the reviewer for their comments. We have revised our descriptions to make them more comprehensive and we have added an additional video to help to distinguish between host and parasite cells (Video 1).

Previous studies in rats and mice have shown that *T. brucei* infection is associated with parasite and T cells extravasation. Did the authors observe similar phenomena with *T. congolense*? Could they document a rupture of the blood brain barrier integrity? Did they observe dextran leakage? Are they sure that the injected anti-CD45 antibody does not diffuse in extravascular compartments due to alteration of the blood brain barrier?

We did not observe any evidence of parasite extravasation either by intravital microscopy or histology. This is consistent with literature (Hornby 1929, 1932; Hornby & Bailey 1930; Fiennes 1952; Losos et al. 1973; Losos & Gwamaka 1973; Banks 1978).

However, we have observed dextran leakage, suggestive of changes in vascular permeability. Therefore, we went back to our images and we quantified vascular permeability in the brain at the first peak of infection with both strains. We have added these data to Figure 2 (panels D-G). We observed that vascular permeability is increased upon infection, but more prominent in the acute rather than chronic strain. We have further shown that it is higher in the posterior parts of the brain, and shows a strong correlation to parasite number (r^2^=0.92 for the acute strain).

We also have evidence of hemorrhages both from histological analysis and intravital microscopy. We have quantified the prevalence of hemorrhages amongst brain vessels (Figure 2F). We found that vessel damage (classified by both the presence of hemorrhages and increased permeability) is more prevalent in the acute strain than in the chronic.

Regarding a potential diffusion of the anti-CD45 antibody into the extravascular compartments, this is highly unlikely. The antibody is only left in circulation for 3 minutes, this protocol has been extensively validated by immunologists as a way to differentiate between intra- and extravascular cells in the brain (Morawski et al., 2017), even in organs where permeability is higher, like the spleen and lungs (Anderson et al. 2012 and 2014). We also show that the vast majority of macrophages, which should mostly be extravascular, is negative for APC-CD45. If this antibody were to diffuse to extravascular compartments, it would stain extravascular/tissue resident macrophages.

However, to further investigate this concern, we compared leakage of anti-ICAM1 antibody in infected and non-infected brain (shown in the table below). We observed no difference in the extravascular anti-ICAM1 Mean Fluorescent Intensity (MFI) between these conditions, suggesting that the antibody does not leak, even though the blood brain barrier is altered. We further tested this by comparing background anti-ICAM1 MFI between intact brain and after the brain is surgically dissected in several parts. Once again, we observed no difference between these conditions, showing that even when the brain is cut in pieces, the antibody that was previously bound to vessels does not leak to the extravascular tissue.

**Author response table 1. sa2table1:** Mean fluorescent intensity of anti-ICAM1 in the brain.

	Intact Brain	Surgically-sectioned Brain	
	*Extravascular*	*Intravascular*	*Extravascular*	*Intravascular*
Non-infected	3.33 ± 0.39	165.45 ± 9.19	2.42 ± 0.15	171.6 ± 9.25
1/148 (acute)	3.12 ± 0.44	243.35 ± 6.18	2.65 ± 0.18	248.00 ± 7.55
IL3000 (chronic)	2.48 ± 0.15	235.40 ± 12.71	2.87 ± 0.27	224.50 ± 13.32

Is there a way to disentangle the contribution of endothelial versus leukocyte ICAM1? In Figure 4F, it seems that ICAM1 increased signal comes from intravascular cells (presumably leukocytes), not endothelial cells.

ICAM1 expression increases in both endothelial cells and intravascular cells (which we agree are leukocytes, and in fact later in the manuscript show that they are monocytes and dendritic cells). In Figure 3E, we quantified ICAM1 expression in endothelial cells only. This was done by manual quantification from the microscopy recordings, using ImageJ. ICAM1 in leukocytes was quantified in Figures 4C, G, J (and found to be mostly DCs and monocytes).

To further address this reviewer’s query, we went back to the images analyzed in Figure 3E, and also quantified ICAM1, ICAM2, and VCAM1 expression from leukocytes. Consistent with our immunophenotyping results, we observed a substantial increase in ICAM1 expression in leukocytes (five-fold more than non-infected). We also observed a modest increase in ICAM2 expression (2-fold). In contrast, VCAM1 was not detected in leukocytes. We have added this data to the manuscript (lines 387-388).

**Author response table 2. sa2table2:** Mean fluorescent intensity of anti-ICAM1, anti-ICAM2, and anti-VCAM1 in the brain vascular endothelium and circulating leukocytes.

	Endothelium		Leukocytes		
	*Non-infected*	*1/148*	*IL3000*	*Non-infected*	*1/148*	*IL3000*
ICAM1	118.96 ± 9.43	144.19 ± 12.61	155.15 ± 14.50	108.18 ± 5.54	504.50 ± 18.33	511.03 ± 17.35
ICAM2	153.59 ± 11.91	143.51 ± 7.72	161.87 ± 8.72	147.48 ± 10.16	336.58 ± 16.59	306.17 ± 15.56
VCAM1	37.99 ± 3.48	42.15 ± 2.92	36.35 ± 3.25	Undetectable	Undetectable	Undetectable

Despite the high expression of ICAM1 phagocytic cells and their vast recruitment to the vasculature, we showed that these immune cells do not cause cerebral disease (Figure 6—figure supplement 1). Therefore, we conclude that the role of ICAM1 in cerebral disease is dependent on its expression by endothelial cells. We have made the distinction clearer in the text (lines 602-604).

In figure 5D the vessels imaged after injection of ICAM1 antibody look very different (smaller) than with the control anti-IgG2. The authors should show similar vessels to allow comparisons.

We have replaced the images in Figure 5D with others showing vessels of similar calibers.

In the in vitro binding experiments (Figure 5 F-G) the authors used parasites isolated from mouse blood. Is it possible that cytoadhered parasites represent only a fraction of the total parasite population, which would be missed by sampling only circulating parasites?

The reviewer raises a very interesting point. It is indeed possible that we are selecting a parasite population that cytoadheres less. It must be said, however, that *T. congolense* cytoadheres extremely well to every surface, including plastic. Currently, we cannot detach the sequestered parasites from vessels and test their sequestration capacity in vitro. We have made this clear in the results (line 514-517) and moved the respective figure to supplementary (Figure 5—figure supplement 1).

Figure 4C is too small, difficult to read.

We have increased the font of this panel.

In videos S2, S3 and S4 the time scale should be provided. In video S2, all cells look non motile, as if there was no flow, why? In video S4, only parasites are visible in the vessel lumen, no other cells (leukocytes, erythrocytes), why?

We have added timescales to all videos.

In video S2 (now video 3), the vessel is clamped, hence the lack of flow. We often observe that when some vessels have large number of parasites, this reduces the overall blood flow in downstream vessels.

In video S4 (now video 7), we see the three parasites, but also one erythrocyte displacing with the flow (1^st^ second). During this video, no leukocytes were recorded. The reduced number of erythrocytes reflects the anemia observed during trypanosomiasis, as well as the fact that this is a small capillary, with slow blood flow.

Reviewer #2 (Recommendations for the authors):*Trypanosoma congolense* is an important animal trypanosome that exhibits significant biological differences to the better studied *Trypanosoma brucei*. Of note, the parasites sequester by binding within the vasculature and organs, one pathological consequence of which can be cerebral trypanosomiasis. In this manuscript the authors identify a strain of *Trypanosoma congolense*, 1/148, that exhibits extreme virulence and cerebral pathology, resulting in mouse death within a few days. This differs from another well-studied strain, IL3000, that generates sustained infections without cerebral pathology.The study initially characterizes the pathology and distribution of the respective *T. congolense* parasite lines, using comprehensive imaging and histology to identify the parasites and their sequestration in different tissues. The appearance of large cerebral lesions is characteristic of *T. congolense* 1/148 although the extent to which these are certain to be the cause of the ultimate death of the animal is perhaps overstated. It was also unclear how representative this acute and drastic cerebral involvement is of infections in the field. In the text, there is also some lack of clarity regarding the distinction between sequestration and cytoadherence. The authors take care to distinguish between these two phenomena (line 142-143) but then apparently use the phrase interchangeably in describing some of their observed parasite distributions on Figure 3 and line 220 forwards. It would be helpful to be more rigorous in the distinction if indeed it is important to understand the respective phenotypes of the parasite lines. Nonetheless, the descriptions of the parasite distribution appear high quality.

We thank the reviewer for their appreciation of our work.

We will refine the text to clarify sequestration and cytoadherence as described in the introduction.

Regarding the representation of acute cerebral trypanosomiasis in endemic settings, we have rewritten the introduction and the discussion to clarify that, currently, it is not common in cattle, but it is reported in other livestock species, as well as dogs, horses, and exotic cattle breeds. We further argue that the introduction of more productive, exotic cattle breeds in Africa would be a major drive for the continent’s economic development. Furthermore, we refer to data that supports that climate change is likely to enlarge the distribution of tsetse flies into temperate climates, which will likely result in an increase in acute disease. Therefore, having an experimental model to address acute trypanosomiasis, and understanding how acute disease can be prevented is of foremost importance to the field.

After characterization of the distribution of the parasites in various tissues, the immune responses to the infection are analyzed by gene expression profiling and then the serum analysis of various cytokines. The elevation of ICAM1 is also associated with the infection although the overall strength of upregulation appears modest. Regardless, the effect of blocking ICAM1 is apparently to reduce the pathology of the infection with 1/148 and associated immune responses are also reduced, providing evidence that the ICAM1 response is disease-relevant. However, in these studies, and more broadly in the analysis of various immune responses, I would have valued a comparison of the IL3000 (non-cerebral) and 1.148 (cerebral) responses rather than infection versus non infection to gain better reassurance of the specificity of responses for the pathological measure under assessment.

We thank the reviewer for the suggestion of including more data on IL3000 to allow a more comprehensive comparison between models. To address this, we have performed the following experiments with the IL3000 strain: quantification of integrin ligand expression in endothelial cells by immunofluorescence analysis, serum cytokine profiling at the first peak of parasitaemia, and immunophenotyping by flow cytometry analysis. These data have now been added to Figure 3E, Figure 4, Figure 3—figure supplement 1, and Figure 4—figure supplement 1. These new data showed that the pro-inflammatory profile in the brain is common to both models of disease, including the increased expression of ICAM1 in the endothelium and the increase in pro-inflammatory cytokines. However, the immunophenotyping experiment clearly showed that the immune responses in both models are very distinct: whilst in the acute (1/148) model we observed increased numbers of intravascular ICAM1^+^ monocytes and extravascular CD4^+^ T cells, in the chronic (IL3000) model, we observed increase in extravascular ICAM1^+^ B cells. We believe these experiments have added considerable value to our work, and so we are grateful for the suggestion.

For the RNAseq study, where we analyze differential gene expression in the brain endothelial cells, we considered comparing IL3000 and 1/148, but we worried that the change in expression profile would not be strong enough to be detected. This is particularly important because IL3000 parasites still populate the brain vasculature in similar levels as 1/148 (even though most as non-sequestered forms), so they are likely to also trigger inflammatory cascades. This was corroborated by the IFA and cytokine profiling results discussed above. Yet, it is our argument that there is a threshold for parasite sequestration before it triggers in an immune response that is so exacerbated that culminates in lethal neuropathology. These subtle changes would possibly not be detected if we compared IL3000 with 1/148 at the transcriptome level. Besides, we no longer have access to RiboTag.PDGFb.iCRE mice, and therefore are unable to perform this experiment with the IL3000 parasite line.

I lack detailed expertise, but it seemed the analysis of several immune responses to 1/148 infection were limited to rather blunt tools in some cases. Where more targeted knock out murine strains are available, chemical inhibition of macrophage using clodronate liposomes was used. Similarly, RAG2 KO cells were used to explore general T cell contributions but this also prevents antibody responses (notwithstanding the acute infection being analyzed where B cells may be less important). Again, more targeted interventions might be possible.

As suggested by both reviewers 1 and 2, we have refined the RAG2 KO model by performing the adoptive cell transfer experiment, now included in the new Figure 7. This has allowed us to identify CD4^+^ T cells as the cause of acute disease, which we believe considerably strengthen our disease model.

Regarding the use of clodronate liposomes to inhibit macrophages, we agree with the reviewer that this method is rather unrefined. However, it is still the best and least damaging option available for depletion of macrophages/monocytes. Macrophages are essential in early stages of embryonic development. So, genetic ablation of macrophages, such as in the CSF1r KO line, are quite harmful for the mice, resulting in progeny with several deficiencies including lack of tooth, osteopetrosis, and reduced BM cellularity (Hua et al. 2018). The latter is a particular concern when studying trypanosomiasis because the disease is known to severely affect hematopoiesis. In contrast, chlodronate liposome ablation of macrophages is fast-acting, very effective, and widely accepted within the immunology community.

It was also intriguing that chronic infections could be sustained in RAG2 KO mice in the absence of antibody control. What do the authors think is happening here? Are the antigenic variation and periodic killing of antigenic variants that characterizes African trypanosomes less important for long term infections than widely considered?

We agree that this is theoretically an unexpected result because B and T cells are expected to be necessary to avoid the host being killed. However, the sustained high parasitemia in RAG2 KO mice has also been previously described in *T. brucei* and shows a similar pattern to what we observed in our work (Machado et al., 2021). The lack of B cells, results in the absence of parasitaemia peaks because there is no antibody clearance. Although the mechanism remains to be studied in *T. brucei* and *T. congolense*, we suspect that parasitaemia is partially controlled by innate, as well as by the fact that parasites themselves regulate their population levels by quorum sensing (MacGregor et al. 2011; Silvester et al. 2017).

In summary, I consider this a valuable model for exploring cerebral trypanosomiasis caused by Trypanosoma congolense. The very acute nature of the infection/pathology is striking and it would be useful to know if this is typical of cerebral trypanosomiasis or another facet of 1/148 that could contribute to some of the responses measured here but less applicable in disease-relevant isolates. The immunological studies are also somewhat preliminary and potentially correlative but reviewers with expertise in this specific area would likely have a better insight into the extent to which causative effects are being studied.

We thank the reviewer for their comments. We hope that our revised manuscript has addressed their concerns regarding relevance and the completeness of immunological studies.

I would be reassured by a more comprehensive comparison between 1/148 and IL3000 to allow a better comparison between cerebral and non-cerebral trypanosomiasis rather than comparisons between infected and uninfected mice. This relates to gene expression studies and the analysis of specific immunological responses.

As explained above, we have performed all requested experiments with the IL3000 strain, with the exception of the RNAseq. This includes quantification of integrin ligand expression in endothelial cells by immunofluorescence analysis, serum cytokine profiling at the first peak of parasitaemia, and immunophenotyping.

I would also prefer to see more specific targeting of immune phenotypes using specific knockout mouse strains where available to improve the resolution of the study compared to ablation of macrophages using liposomal killing, or quite generic RAG2 KO lines, for example.

This comment was also addressed above.

Reviewer #3 (Recommendations for the authors):This is the first time a detailed model for *T. congolense* brain pathology is being reported. The results are interesting, and a set of very well executed detailed experiments show the involvement of the ICAM1-CXCL9/10-IFNgamma- T cell axis. This is an interesting observation as it mirrors findings of the *T. brucei* models for human trypanosomiasis, with the big difference that the latter involves infiltration of the parasite into the brain parenchyma, while *T. congolense* remains in the microvasculature.Despite the strengths of the experimental procedures, and the vast amount of data provided, it is not clear to which extent this unique mouse model is relevant for the in vivo field situation of *T. congolense* infections in cattle. In the case of the latter life-threatening pathology relates mostly to anaemia and a metabolic wasting disorder, rather than neuropathology. In that aspect, while interesting from a fundamental point of view, the paper might not shed new light on the true nature of animal trypanosomosis. In this context, attention should be given to the references used to support the potential role of neuropathology in AT: Tuntasuva et al. 1997 is a *T. evansi* paper (a very different parasite closely related to *T. brucei*, and not *T. congolense*), and so are the Rodriguez 2009 paper and the Busher 2019 paper, dealing with equine trypanosomosis caused by *T. evansi* and *T. equiperdum*. Finally, the Harrus 1995 paper is indeed a *T. congolense* case description of 2 dogs but is not a reference to livestock disease. Hence, none of the references support the idea that cattle or livestock trypanosomosis (the true problem of *T. congolense*) is hallmarked by neuropathology. For clarity towards the reading public, this section of the introduction should provide a more accurate description.

We thank the reviewer for this comment and we acknowledge that have improved the introduction to clarify the current contribution of neuropathology to *T. congolense* epidemiology.

The second fundamental concern that requires a detailed discussion/consideration is the fact that only the rather unique acute 1/148 model studied in this paper seems to result in marked brain pathology in ice. In contrast, the well-described IL3000 model does not exhibit this feature. Taken than in the field T. congolense is a chronic infection, once again the question arises as to how relevant the observations of the acute rapid killing mode are.

The relevance of the acute disease model to endemic settings is argued above and has been included in the manuscript introduction and discussion.

Finally, the authors speculate on why they assume brain pathology is the cause of death of the infected mice, they acknowledge the occurrence of acute kidney disease at the time of death as ell. Hence, it is hard to see how the latter can be ignored as a cause of death as kidney failure has been described in other experimental trypanosome models and can result in t in sudden death within hours.

Brain pathology was considered the most likely cause of death for two reasons: severe brain pathology and the fact that mice present severe neurological impairment close to the time of death. However, we agree that understanding the contribution of kidney damage to acute disease is an important aspect.

To address this, we have performed a biochemical analysis of urea, creatinine, and neutrophil gelatinase-associated lipocalin (NGAL) from serum of mice infected with either 1/148 or IL3000 at the first peak of parasitaemia. Urea allow us to detect dehydration, creatinine is a standard indicator of kidney function, whereas NGAL is an early indicator of acute kidney injury (Soni et al. 2010, PMID: 19582588). Our results, now included in Figure 1F, show signs of acute kidney injury in the 1/148-infection only (as indicated by elevated NGAL levels), but still uncompromised kidney function in both models of disease (as suggested by normal creatinine levels). Elevated levels of urea, in the context of normal kidney function, are suggestive of mild dehydration and apply to both models. These results clearly show that so kidney failure is not the most likely cause of death. These results have been described in Results section, lines 139-149.

Furthermore, we have strengthened our pathology results by conducting an additional experiment to increase the N, showing individual values, rather mean values, as well as p-values to assess differences between lesion scores in each organ (Figure 1C).

Overall, this is a very impressive work and it shows how modern techniques in experimental immunology can be combined to get a comprehensive vision on the development of infection-associated multi-organ pathology. It remains to be seen if this impressive knowledge truly relates to the disease of livestock trypanosomosis.This is an impressive paper and the execution of the work is done at a level where this reviewer feels it would be almost unfair to request any additional experiments. The only concern is that the results might 'only' explain one unique condition of acute mouse T. congolese trypanosomosis. There are many models for T. congolense described in literature, and virtually all of them are characterized by multiple parasitemia peaks. And also in the field, T. congolense livestock infections are chronic wasting diseases, not acute neuropthological diseases with a quick deadly outcome. That would be a typical problem for T. equiperdum, but not T. congolense. The open discussion about this issue is missing from the paper. This could be improved, as the data itself is valuable by istelf, even if they do not represent the general pathology of livestock T. congolense AT.

We thank the reviewer for their comments. We hope that our revised manuscript has addressed their concerns regarding relevance of acute trypanosomiasis.

[Editors’ note: what follows is the authors’ response to the second round of review.]

The manuscript has been improved but there are some remaining issues that need to be addressed, as outlined below:– We kindly ask you to tone down the stated general biological importance of the acute model and the risk the acute pathology generates in wider geographical areas (see comments of reviewer #2).

We would like to thank the reviewers for their positive assessment of our work. We have addressed all of their final concerns.

Reviewer #2 (Recommendations for the authors):The authors have made very substantial changes to the presented manuscript- particularly by increasing the comparative analysis between T. congolense isolates causing chronic (IL3000) versus acute (1/148) disease with cerebral involvement. These comparisons have been informative and addressed my main experimental comments from the last submission, further revealing the contribution of CD4^+^ T cells to the pathological phenotype. I am satisfied with the changes made in the experimental work given the constraints and what is possible (e.g., the RNAseq experiments could not be extended), though a reviewer with greater immunological expertise will provide better insight.I remain a little hesitant on the relevance and value of the model with respect to T. congolense infection and pathology however. The authors' text previously mis-stated the importance of cerebral disease in livestock infections for T. congolense. In the new introduction they have revised this to reflect the evidence that other species (dogs, horses, goats, productive Bos taurus breeds) can suffer acute disease, and invoke the 1/148 model as a tool to address this. However, it seems the acute pathology is a consequence of the host and not the parasite in these instances, and so an alternative animal model rather than an additional parasite model such as 1/148 would be the more useful tool.The risk of tsetse range expansion to threaten livestock susceptible to acute *T. congolense* infection also seems overstated. Unlike many hematophagous flies- e.g., mosquitos or sandflies- the biology of tsetse limits their capacity to significantly increase their range and evidence to date suggests that climate change has rather had the opposite effect. For other African trypanosomes (*T. vivax* and *T. brucei*) spread outside the tsetse belt has been possible by mechanical transmission but I am not aware of evidence supporting persistent and sustained spread of *T. congolense* by this mechanism (despite there being one report of inefficient spread by tabanids). Hence the potential risk to susceptible breeds in other geographical settings may be very limited.Overall, therefore, I applaud the additional experimental data provided which usefully enhances the paper from its original form and will be of broad interest, I think. I do retain some doubts over the general value of the 1/148 model, however and some of the wider biological implications of the work which potentially remain limited.

We thank the reviewer for their positive assessment of our work.

We agree with this reviewer that the outcome of the disease will depend on the pair animal species + parasite strain. In the papers describing acute disease in dogs, horses, goats the parasite strains were not identified and thus we do not know how different they are from 1/148 and IL3000. The specific contributions of parasite and host factors and their synergies is a very important question that awaits to be answered by several groups in the field.

To tone down the general importance of the acute model, we have rephrased the final paragraph of the discussion. We have also completely removed the paragraph in the discussion mentioning the potential spread of T. congolense to current temperate regions due to climate change (lines 574-594). We have checked that we do not mention this hypothesis anywhere else in the manuscript.

Reviewer #3 (Recommendations for the authors):This reviewer would like to thank the authors for addressing the concerns posted in the previous round of reviewing in a very thorough manner. Addressing the kidney and the CD4 T cell issue was done by a convincing experimental approach, and the discussion on whether or not cerebral complications are a major contributor to T. congolense livestock death is now addressed by a more balanced phrasing as well as new and recent references. Hence, all major comments put forward by the reviewer in the past have been addressed and the paper has become an even more impressive piece of work. It will probably inspire other researchers in the field of AT to pay more attention to this pathology aspect.

We thank the reviewer for their positive assessment of our work.